# A digital single-molecule nanopillar SERS platform for predicting and monitoring immune toxicities in immunotherapy

Junrong Li [1], Alain Wuethrich [1✉], Abu A. I. Sina [1], Han-Hao Cheng [2], Yuling Wang [3✉], Andreas Behren [4,5], Paul N. Mainwaring[1] & Matt Trau [1,6✉]

The introduction of immune checkpoint inhibitors has demonstrated significant improvements in survival for subsets of cancer patients. However, they carry significant and sometimes life-threatening toxicities. Prompt prediction and monitoring of immune toxicities have the potential to maximise the benefits of immune checkpoint therapy. Herein, we develop a digital nanopillar SERS platform that achieves real-time single cytokine counting and enables dynamic tracking of immune toxicities in cancer patients receiving immune checkpoint inhibitor treatment - broader applications are anticipated in other disease indications. By analysing four prospective cytokine biomarkers that initiate inflammatory responses, the digital nanopillar SERS assay achieves both highly specific and highly sensitive cytokine detection down to attomolar level. Significantly, we report the capability of the assay to longitudinally monitor 10 melanoma patients during immune inhibitor blockade treatment. Here, we show that elevated cytokine concentrations predict for higher risk of developing severe immune toxicities in our pilot cohort of patients.

[1] Centre for Personalised Nanomedicine, Australian Institute for Bioengineering and Nanotechnology (AIBN), The University of Queensland, Brisbane, QLD, Australia. [2] Centre for Microscopy and Microanalysis, The University of Queensland, Brisbane, QLD, Australia. [3] Department of Molecular Sciences, ARC Centre of Excellence for Nanoscale BioPhotonics, Faculty of Science and Engineering, Macquarie University, Sydney, NSW, Australia. [4] Oliva Newton-John Cancer Research Institute, School of Cancer Medicine, La Trobe University, Heidelberg, VIC, Australia. [5] Department of Medicine, University of Melbourne, Heidelberg, VIC, Australia. [6] School of Chemistry and Molecular Biosciences, The University of Queensland, Brisbane, QLD, Australia. ✉email: a.wuethrich@uq.edu.au; yuling.wang@mq.edu.au; m.trau@uq.edu.au

The advent of immune checkpoint therapy has revolutionised the landscape of traditional cancer treatment and is believed to constitute the backbone of managing certain malignancies[1–3]. By capitalising on the blockade of immune checkpoint inhibitors to take the brakes off parts of the immune system, this emerging therapy has achieved great success producing long-lasting responses (e.g., 10 years or more) in a small but significant fraction of patients[3–6]. Nevertheless, upon the blockade of immune checkpoint molecules, the activated and potentiated immune reaction predisposes patients to a significant risk of immune-related adverse events (irAEs), which can occur in up to 80% of patients receiving immune checkpoint therapy[7–9]. The high incidence of irAEs, which may manifest at any time during treatment, can offset the clinical benefits, lead to premature therapy cessation, and even be life-threatening for certain patients[10–12]. To assist the successful implementation of immune checkpoint therapy, the use of predictive biomarkers for early identification and vigilant monitoring of irAEs is thus critical and a pressing need in avoiding or ameliorating detrimental effects and adjusting therapeutic options.

Cytokines, small signalling proteins, are promising candidates to indicate the occurrence of irAEs due to their prominent role in modulating the anti-cancer immune responses, including enhancing antigen priming, recruiting immune cells into the tumour microenvironment, and upregulating certain immune checkpoint molecules[9,13,14]. Particularly, excessive cytokine secretion has been implicated in severe inflammation as a major constituent leading to irAEs. For example, the overproduction of fibroblast growth factor 2 (FGF-2)[15–18], granulocyte colony-stimulating factor (G-CSF)[19], granulocyte-macrophage colony-stimulating factor (GM-CSF)[20], and fractalkine (CX3CL1)[21] have been found to participate in immune-related inflammatory disease (e.g., rheumatoid arthritis, autoimmune gastritis, and Crohn's disease). These inflammatory cytokines have recently been reported to indicate irAEs for melanoma patients who underwent immune checkpoint therapy[9]. The clinical deployment of cytokine analysis for irAE monitoring is challenging and requires a technology that can (i) determine the selected cytokines with great sensitivity[9], especially at the onset of irAE development, where the cytokine concentrations are likely to be the lowest; as well as (ii) simultaneously detect a panel of cytokines to reflect the complex interplay of cytokine signalling pathways[22] and the variable irAE symptoms among patients.

Conventional cytokine analyses such as immunosorbent assays have limited clinical applicability for irAE assessment due to their limited capacity to detect low cytokine concentrations in blood as well as for assessing a panel of cytokines in a single sample simultaneously. Recently, advances in micro/nanomaterial-based systems have provided a promising suite of technologies that improve the conventional assays by overcoming the above limitations[23,24]. Encouragingly, the unique advantages of micro/nanomaterial-based systems convey an attractive option for cytokine analysis with the desired results of high sensitivity and multiplexing. The high specific surface area of these miniaturised materials increases mass transfer subsequently enhancing the interaction with target molecules and thus improving the detection sensitivity[25]. The capabilities of micro/nanomaterial fabrication techniques permit individually separated compartments sufficiently discrete to hold single molecules and hence encompasses a promising strategy for counting assays that can further push the sensitivity of the traditional assays[24,26,27]. Moreover, the physicochemical properties of nanostructured materials can be exploited to simultaneously label multiple targets (e.g., various spectral signatures) for high-throughput parallel measurements[28–30]. Therefore, by combining the potential of micro/nanomaterial systems with the need for sensitive irAE monitoring, we have developed a platform for sensitive and multiplex cytokine counting analysis.

Combining the use of (a) discrete single cytokine nanopillar array chip with discretely separated compartments, (b) control of target concentrations to follow a Poisson distribution, and (c) the recognition of target by single-particle active surface-enhanced Raman scattering (SERS) nanotags with a confocal Raman microscope allows accurate and in situ counting of a multi-cytokine panel (FGF-2, G-CSF, GM-CSF, and CX3CL1). Different from the fluorescence-based digital counting strategies[24,26], the strikingly narrow spectral peaks of SERS (~1–2 nm) in comparison to fluorescence (~50 nm) makes this platform intrinsically ideal for multiplexed cytokine analysis[28].

In this work, we present a digital nanopillar SERS platform that enables the specific cytokine quantification down to attomolar levels and the application in melanoma patients receiving immune checkpoint blockade therapy. Beyond the capability to predict irAEs in melanoma patients receiving therapy, the digital nanopillar SERS assay could potentially be extended to other cytokine-associated immune responses such as excessive immune activation due to viral or bacterial infections (such as COVID-19).

## Results

### Digital nanopillar SERS platform for parallel profiling of single cytokine.
Our concept of digital nanopillar SERS platform for cytokine analysis relies on Rayleigh criterion separation, probability-driven Poisson distribution, single-particle active SERS nanotags, and confocal SERS mapping (Fig. 1). To precisely fabricate the pillar array, we opted to use an electron beam lithographic approach to write the array into a photon-sensitive material followed by physical vapour deposition of gold to create the gold-topped pillars, and selectively reactive ion etching to reveal the pillar structure (Supplementary Fig. 1). The nanopillar array chip consisted of 250,000 individual pillars. As shown in the scanning electron microscope (SEM) image of Fig. 1a, the cubic nanopillars have an edge-to-edge width of 1000 nm and are evenly distributed at 1000 nm intervals to suit the lateral Raman microscope resolution (~1000 nm) that fulfils the Rayleigh criterion separation required to acquire a single SERS spectrum from each pillar without spectral overlap from adjacent pillars.

By using specific gold-thiol chemistry with the linker molecule dithiobis (succinimidyl propionate) (DSP), the gold-topped pillars were selectively functionalised with target recognition antibodies (anti-FGF-2, anti-G-CSF, anti-GM-CSF, and anti-CX3CL1) and acted as the small compartments to capture and confine the individual cytokine. Upon DSP binding on the gold-topped pillars through gold-thiol bond, DSP uses N-hydroxy-succimide (NHS) ester to react with the amine groups of the antibodies[31,32]. The successful antibody conjugation on gold-topped pillar surfaces was confirmed by matrix-assisted laser desorption ionisation-time of flight mass spectrometry (MALDI-TOF MS) (Supplementary Fig. 2), which showed high molecular weight fragments derived from antibodies. Furthermore, spectroscopic ellipsometry was utilised to estimate the antibody density on pillar surfaces. Based on the obtained film thickness of 18.5 nm, the calculated antibody surface density was 5.5 mg/m² using the Cuypers model[33], which was in agreement with the reported antibody density on substrate surfaces[34]. Though these characterisations indicated the presence of antibodies on the pillar array, it was not possible to assess the exact distribution of the four structurally related (same immunoglobulin G family) antibodies on a single pillar with an area of 1 μm². As an advantage of the digital read-out with a large redundancy of pillars, it is not essential to have all four types of antibodies equally distributed on a single pillar for the assay to work. The

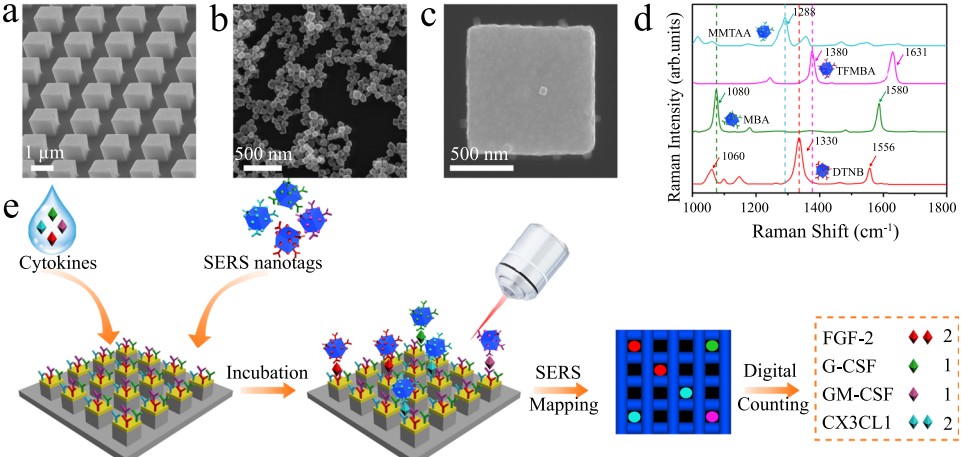

**Fig. 1 Digital single-molecule nanopillar surface-enhanced Raman scattering (SERS) platform for parallel counting of four types of cytokines.** SEM images of **a** pillar array side view, **b** nanoboxes, and **c** a single nanobox on the top of a pillar; **d** SERS spectra of nanoboxes conjugated with 5,5-dithiobis (2-nitrobenzoic acid) (DTNB), 4-mercaptobenzoic acid (MBA), 2,3,5,6-tetrafluoro-4-mercaptobenzoic acid (TFMBA), or 2-mercapto-4-methyl-5-thiazoleacetic acid (MMTAA) Raman reporters; **e** workflow for multiplex counting of cytokines, including fibroblast growth factor 2 (FGF-2), granulocyte colony-stimulating factor (G-CSF), granulocyte-macrophage colony-stimulating factor (GM-CSF), and fractalkine (CX3CL1). Data from one independent experiment.

combined surface area of all antibody-conjugated pillars provides an excess of cytokine binding sites, which maximises successful cytokine capture within the pillar array. Supplementary Fig. 3 shows the SERS mapping images of an equimolar cytokine solution (1031 aM) that provided a similar signal count for the FGF-2, GM-CSF, G-CSF, and CX3CL1 SERS nanotags, indicating a required distribution of four kinds of antibodies conjugated to the array of pillars. Instrumental to the digital counting of cytokines, we controlled the target concentration based on the principle of Poisson distribution where the ratio of cytokine molecules to pillar number was <1:10, ensuring a 99% probability that there was either one cytokine molecule or zero per pillar[24]. At a ratio of 1:10, 10% of all pillars were occupied or activated with the cytokine molecules.

Following the capture of cytokines on nanopillars, SERS nanotags were applied to recognise the captured cytokines. The preparation of SERS nanotags was performed by the co-conjugating of Raman reporter and target antibody onto gold–silver alloy nanoboxes. Specifically, an average size of 80 nm gold–silver alloy nanoboxes were firstly synthesised using a rapid and aqueous phase approach[35] as indicated in the SEM image in Fig. 1b. Supplementary Fig. 4a, b shows the transmission electron microscope (TEM) image of the nanoboxes with the hollow inner structure and a wall thickness of around 15 nm. Nanoparticle tracking analysis (NTA), which allows the tracking and detection of single particles, shows the nanoboxes have a mode size of 77 nm (D10 = 67.6 nm and D90 = 110.6 nm) (Supplementary Fig. 4c). UV-vis extinction spectroscopy demonstrates the nanoboxes possess a surface plasmon resonance (SPR) peak at 610 nm (Supplementary Fig. 4d). The resonance frequency of the nanoboxes enables a more sensitive signal readout with 632.8 nm laser excitation[29], which also has a higher Raman scattering efficiency than 785 nm laser. Thereafter, four types of Raman reporters (5,5-dithiobis (2-nitrobenzoic acid) (DTNB), 4-mercaptobenzoic acid (MBA), 2,3,5,6-tetrafluoro-4-mercaptobenzoic acid (TFMBA), and 2-mercapto-4-methyl-5-thiazoleacetic acid (MMTAA)) that generate unique Raman signals (1330 cm⁻¹, 1080 cm⁻¹, 1380 cm⁻¹, and 1288 cm⁻¹) were coupled with their corresponding detection antibodies onto nanoboxes as specific SERS nanotags for identification of FGF-2, G-CSF, GM-CSF, and CX3CL1, respectively. As shown in

Fig. 1d, the four SERS nanotags provide the strong and non-overlapping Raman signals, which facilitates the multiplexing analysis of four cytokines. The assignment of the major Raman peaks from the four Raman reporters was summarised into Supplementary Table 1. To evaluate the SERS enhancement property of the nanoboxes, we calculated the enhancement factor (EF) of the four Raman reporters on the nanoboxes. Based on the labelled characteristic peaks in Supplementary Fig. 5, the calculated EFs of DTNB, MBA, TFMBA, and MMTAA were $8.14 \times 10^6$, $1.46 \times 10^7$, $4.01 \times 10^7$, and $3.26 \times 10^7$, respectively. The obtained EFs were higher than the reported spherical gold nanoparticles and pure silver nanocubes[36] and comparable to the reported hollow nanocubes[37], illustrating the high SERS property of the nanoboxes. To investigate the SERS nanotag stability, we monitored the Raman signal intensity over 7 days. As shown in Supplementary Fig. 6, Raman signal intensity variations are less than 5% in the SERS spectra, suggesting the good stability of the prepared SERS nanotags.

The following SERS mapping generated false-colour images for counting single cytokine molecules. Under the Raman microscope, the pillar array was visualised as a blue and black grid by representing the specific Raman shifts corresponding to the silicon signals (520 cm⁻¹), in which the blue colour was assigned to silicon signals showing silicon substrates and the black colour indicated the gold-topped pillars because of the lack of silicon signals. The representation of the four colours of the SERS nanotags (red, green, purple, and cyan) on the gold-topped pillars (i.e., black) reflected cytokine molecule occupation (FGF-2, G-CSF, GM-CSF, and CX3CL1). Elevating the sensing area (or gold-topped pillars) from the silicon substrate was selected as a strategy to minimise the false-positive events. By using the confocal function of the Raman microscope, the laser was selectively focused on the gold-topped pillars, thus largely removing the background signals from potentially non-specifically adsorbed SERS nanotags on the silicon substrate. Finally, the specific SERS nanotag signals present or absent on the gold-topped pillars were counted and represented as percentage of active pillars used for total cytokine quantification. For statistical calculations, SERS mapping was applied for scanning 6480 pillars. This digital counting mode, therefore, has the potential to reach the ultimate sensitivity of single molecule cytokine detection.

**Demonstration of the single-particle SERS activity of gold–silver alloy nanoboxes.** The successful implementation of digital nanopillar SERS assay necessitates the use of single-particle active plasmonic nanostructures that give a clearly detectable signal for each of the single cytokine binding event. The single-particle SERS detection sensitivity is essential to this assay development as the single-particle inactive plasmonic nano-particles (e.g., spherical gold nanoparticles)[38] would unavoidably result in an underestimate of cytokine concentration.

We evaluated the single-particle SERS activity of the prepared anisotropic nanoboxes by acquiring the signals from the individual nanoboxes that were labelled with DTNB reporters. The use of DTNB, a non-resonant Raman reporter, guaranteed the Raman signal enhancement was solely contributed from the nanobox-generated electromagnetic field. As seen in the SEM image (Fig. 2a), two clearly separated DTNB-labelled nanoboxes were deposited on the silicon wafer (highlighted in red circles). The corresponding Raman image (Fig. 2b) displayed several bright Raman spots originating from these individual nanoboxes. The elongated bright Raman spots in the SERS mapping image were probably caused by the slight aggregation of several nanoboxes during sample preparation processes (e.g., centrifuga-tion)[38], which was difficult to visually resolve in the SEM image (Fig. 2a). However, unlike the intensity-based assay, the aggregated nanoboxes as SERS nanotags to target cytokine will not skew the digital readout result, because each cytokine will occupy a single pillar following Poisson distribution and both aggregated and individual nanoparticles are regarded as a single binding event that truly reflects the target number[39]. We then acquired the SERS spectra from two individual SERS nanoboxes (Fig. 2c, (1) and (2)) and two separate spots of bare silicon (Fig. 2c, (3) and (4)). The presence of nanoboxes showed the characteristic Raman signal at 1330 cm$^{-1}$ from DTNB, whereas the silicon spectra (3) and (4) lacked the specific peak. This observation demonstrated the single-particle SERS activity of nanoboxes, which was largely attributed to the enhanced electromagnetic fields of nanoboxes on specific regions (e.g., tips and corners)[40,41] and thereby facilitated the sensitive and accurate counting of cytokines. Based on the acquired SERS mapping image, the median (interquartile range) of the DTNB peak intensity (1330 cm$^{-1}$) in the presence and absence of nanoboxes were 183.03 a.u. (149.48–243.35 a.u.) and 18.07 a.u. (15.51–23.12 a.u.), respectively. Furthermore, the mean ± standard deviation of the DTNB peak intensity with nanoboxes (213.41 ± 85.03 a.u.) distinguished clearly from the position without nanoboxes (18.79 ± 6.01 a.u.), which demonstrated the feasibility of correctly identifying the presence of nanoboxes.

**Optimisation of digital nanopillar SERS platform for cytokine detection.** The reliable detection of single cytokine molecules by the digital nanopillar SERS platform depends on the geometric features of the pillar array (i.e., pillar height, cross-section area of pillar) and assay conditions (i.e., incubation time for sample and SERS nanotags).

We first sought to investigate the effect of pillar height on Raman signal intensity to differentiate signals from non-specifically bound SERS nanotags on the silicon substrate and specifically bound SERS nanotags on the gold-topped pillars. FGF-2 SERS nanotags were randomly deposited on the silicon substrate to mimic the non-specific binding scenario and the Raman mappings were perfomed by moving the objective along the z-axis direction with different heights (0 nm, 500 nm, 1000 nm, and 1500 nm) to compare the signal intensity. In Fig. 3, a–d show the false-colour SERS images and e, f the corresponding SERS spectra with characteristic DTNB reporter peak at 1330 cm$^{-1}$ acquired from the circled spots in a–d. At the height of 0 nm where the SERS nanotags were in focus, we noticed bright Raman spots (Fig. 3a) and strong Raman signals (black line in Fig. 3e, f). With increasing z heights to 500 nm and 1000 nm, the Raman spots

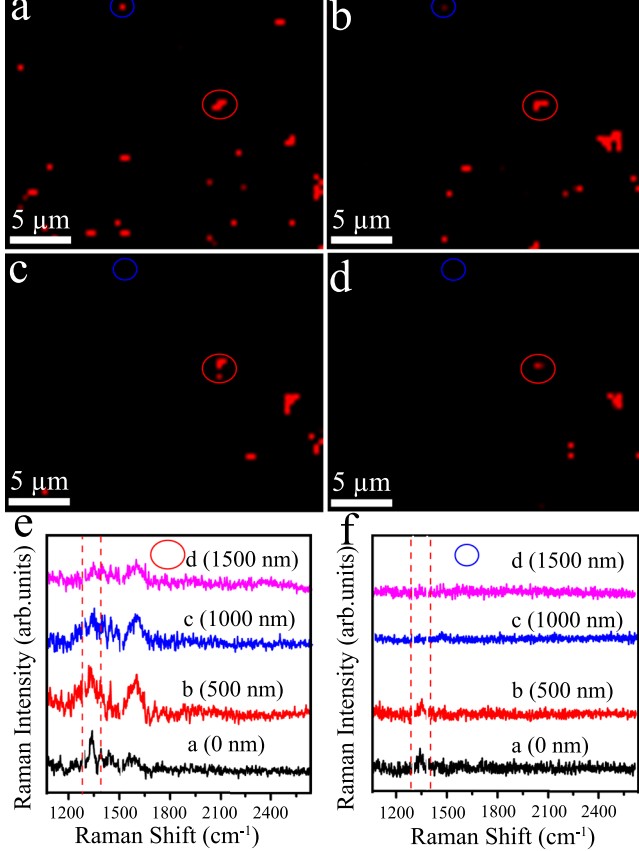

**Fig. 3 Study of confocal height on Raman signal intensity.** SERS mapping of FGF-2 SERS nanotags on the silicon substrate with changing confocal height of **a** 0 nm, **b** 500 nm, **c** 1000 nm, and **d** 1500 nm; selected Raman spectra obtained from **e** red circles and **f** blue circles of SERS images. Red dotted lines in **e** and **f** indicate peak signal at 1330 cm$^{-1}$ from DTNB. Data from one independent experiment. Source data are provided in the Source Data file.

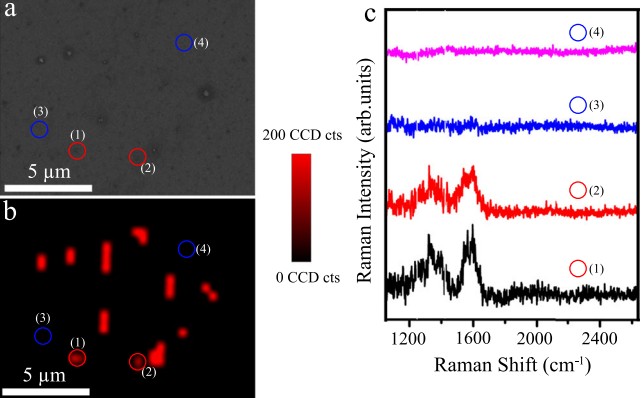

**Fig. 2 Demonstration of the single-particle SERS activity of DTNB-labelled nanoboxes. a** SEM image and **b** corresponding SERS mapping image of DTNB-labelled nanoboxes on a silicon substrate; **c** representative SERS spectra of numbered locations indicated in **a** and **b**. The red dotted line shows the characteristic peak at 1330 cm$^{-1}$ from DTNB. Data from one independent experiment. Source data are provided in the Source Data file.

decreased (Fig. 3b, c) and the signal intensity weakened/disappeared (red and blue lines in Fig. 3e, f) as the nanoboxes became increasingly out of focus. A further increase to 1500 nm did not remarkably weaken Raman signals compared to the height of 1000 nm (Fig. 3d). Hence, for the fabrication of the pillar array chip, we selected a pillar height of 1000 nm to greatly reduce the potential interference from non-specific signals.

The cross-section area of the pillars provides the space for cytokine binding and labelling with SERS nanotags. To study the effect of pillar cross-section area, we fabricated array chips with pillars of various widths (250 nm, 500 nm, and 1000 nm) (Supplementary Fig. 7a–c) and functionalised with anti-FGF-2 antibody. Each chip consisted of 250,000 individual pillars. We then analysed a sample that contained ~25,000 molecules of FGF-2 (40 μL, 1031 aM), which should result in 10% active pillars (ratio FGF-2: pillars of 0.1). As seen in Supplementary Fig. 7d–f, an increasing pillar cross-section area results in a higher fraction of active pillars. In reference to the expected active pillar percentage (10%), the 250 nm and 500 nm pillar arrays produced lower active pillars (2% and 6%), which suggested a significant loss of target recognition by SERS nanotags. For the 1000 nm wide pillars, the active pillar percentage was 11%, close to the nominal value of 10%. We further tested a sample with 260 aM FGF-2 (i.e., 2.5% active pillars) on the pillar array chips with 250, 500, and 1000 nm pillar widths. The capture efficiency of these three chips was summarised in Supplementary Table 2. In comparison to the pillar array of 250 nm and 500 nm sizes, the 1000 nm provided an improved capture efficiency. As the accessible target recognition surface area per pillar increases, it can possibly promote the thermodynamics and kinetics for higher surface binding and capture efficiency[25]. Consequently, the 1000 nm pillar array was adopted in the subsequent experiments.

An optimal incubation time of cytokine and SERS nanotags on the pillar array can shorten the operation time and reduce the potential risk of nonspecific binding that could lead to false-positive counting. We thus studied the effect of incubation time of cytokine with SERS nanotags for 30 to 90 min in a solution of 1031 aM FGF-2 and FGF-2 SERS nanotags. As suggested by Supplementary Fig. 8, the increase in incubation time gives rise to a higher proportion of active pillars. In comparison with the theoretical active pillar percentage (10%), both 30 min and 60 min incubation time were able to provide a desirable active pillar percentage (11% and 13%, respectively). A longer incubation time (90 min), however, reported an active pillar percentage (20%) two times higher than the theoretical, indicating the occurrence of nonspecific absorption of SERS nanotags on the pillar array chip. Thus, we selected 30 min incubation time for further digital nanopillar SERS measurements.

**Specificity of the digital nanopillar SERS platform for cytokine detection.** Accurate and reliable recognition of the specific target is essential for cytokine quantification in clinical samples. To demonstrate the detection specificity of the digital nanopillar SERS assay, we prepared an anti-FGF-2 antibody functionalised pillar array and measured samples containing target FGF-2 cytokine and controls (G-CSF, GM-CSF, CX3CL1, and PBS). It was observed that only the presence of FGF-2 activated significant amounts of pillars whereas the negative controls only generated negligible active pillars (Fig. 4), indicating the high specificity for FGF-2 detection. Similarly, we studied the specific detection of G-CSF, GM-CSF, and CX3CL1, as shown in Supplementary Figs. 9–11, in which the typical Raman images displayed high proportions of active pillars in the presence of specific targets but not for the negative controls.

To further investigate the specificity of binding between SERS nanotag and antibody-functionalised pillar, we performed SEM analysis to "closely" inspect the pillar array for the presence or absence of SERS nanotags. As a representative model, we selected to image FGF-2 SERS nanotags on the anti-FGF-2-functionalised pillar array after sample incubation with FGF-2 cytokine and non-target controls (Fig. 5). As expected, we observed the cubic nanoparticles on the top of pillars in the presence of FGF-2 due to the successful recognition of SERS nanotags. On the contrary, pillar arrays did not display a significant number of FGF-2 SERS nanotags with non-target controls. Consequently, the consistent Raman and SEM data demonstrate the capability of the assay for specific target cytokine counting. The ability to selectively identify these four cytokines in the designed assay is critically important for their usage in clinical samples.

**Sensitivity of the digital nanopillar SERS platform for cytokine detection.** As there is typically low abundance of cytokines in

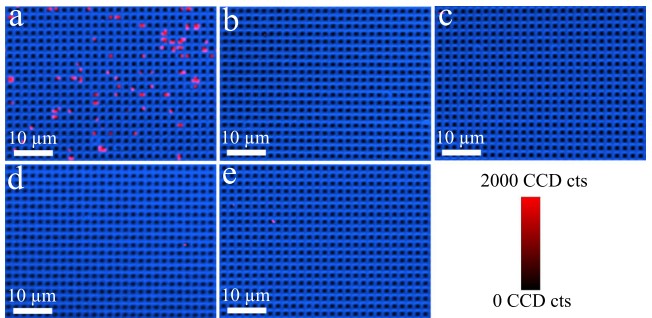

**Fig. 4 Specificity of digital nanopillar SERS platform for FGF-2 cytokine detection.** Representative confocal SERS images in the presence of **a** target FGF-2 (1031 aM), and negative controls with non-target controls **b** G-CSF (1031 aM), **c** GM-CSF (1031 aM), **d** CX3CL1 (1031 aM), and **e** PBS. The median (interquartile range) of active pillars per scanning image for FGF-2, G-CSF, GM-CSF, CX3CL1, and PBS was 72 (63.5–76.75), 1.5 (1.5–2), 2 (1–4), 0.5 (0–1.25), and 1 (1–1.75), respectively. Data from one independent experiment.

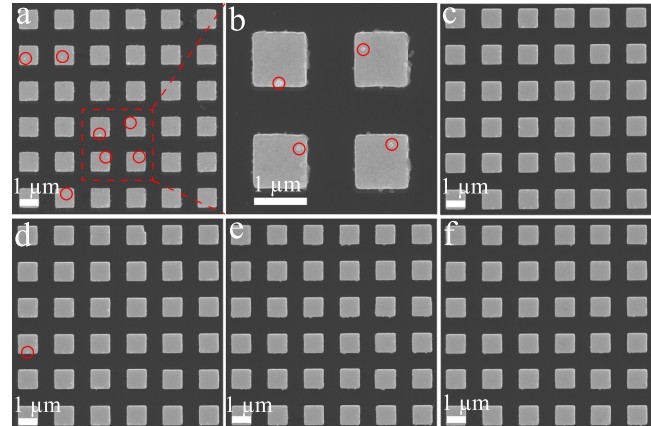

**Fig. 5 Specificity of the digital nanopillar SERS platform for FGF-2 cytokine detection.** Representative SEM images of pillar array incubated with FGF-2 SERS nanotags in the presence of **a, b** FGF-2 (1031 aM), **c** G-CSF (1031 aM), **d** GM-CSF (1031 aM), **e** CX3CL1 (1031 aM), and **f** PBS. The red circles highlight the existence of SERS nanotags. Panel **b** is the magnified SEM image of the red-highlighted section in **a**. It is noted that nanofabrication debris on the sidewall of the pillars can also be seen. Data from one independent experiment.

clinical samples, the technique based on cytokine detection is expected to possess sufficient sensitivity to reliably assess irAEs[9]. To investigate the sensitivity and dynamic detection range of the digital nanopillar SERS assay, we firstly titrated the designated concentration of one target cytokine (FGF-2) on the pillar array chip with 250,000 pillars. To comply with the Poisson distribution, the upper number of cytokine molecules in the sample is 25,000 which should result in 10% activated pillars. Based on this upper molecule number, we were motivated to challenge the assay by serially diluting the number of cytokine molecule in the sample from 25,000 (1031 aM), 6305 (260 aM), 631 (26 aM), and 63 (2.6 aM). As suggested by the Raman images in Supplementary Fig. 12 and with the decrease in FGF-2 molecules, the percentage of active pillars decreased correspondingly from 9.39% for 1031 aM, 6.59% for 260 aM, 1.12% for 26 aM, and 0.62% for 2.6 aM, showing a strong correlation that facilitates quantitative cytokine analysis.

Subsequently, we were interested in exploring the multiplexing capability of SERS to investigate the digital nanopillar SERS assay's dynamic range for the simultaneous quantification of all studied cytokines. As the targets independently follow Poisson distribution, each of the cytokine was separately controlled to activate less than 10% pillars. The specific SERS nanotags provided unique signals for each cytokine that was visualised in the false-colour SERS images by a different colour, thereby enabling in situ and simultaneous cytokine detection. As suggested by the confocal SERS images in Fig. 6, an increase in cytokine concentration corresponded with a higher percentage of active pillars. To facilitate quantitative measurements of the cytokines, we calculated the logarithmic transformation of the percentage of active pillars versus cytokine concentration (Supplementary Fig. 13) confirming the strong statistical and potentially clinically relevant correlation (coefficient of determination ($R^2$) >0.97) observed in the SERS images.

To further investigate the multiplexing quantification performance of the digital nanopillar SERS assay in human serum, we spiked standard cytokines in human serum and tested the dynamic range. Supplementary Fig. 14 shows the linear relationship curves for the four targets. Because of the more complicated sample matrix composition in human samples, the lowest detectable cytokine concentration (5.2 aM) was higher than the PBS solution (2.6 aM).

At a cytokine to pillar ratio of 1:10, we studied the probability of each pillar being occupied by different molecule numbers. To experimentally investigate the number of molecules on a single pillar, we analysed a cytokine mixture that contained all four target cytokines at equal concentration (i.e., ~6250 molecules per cytokine). To visualise and count molecule binding events on a single pillar, we labelled the captured cytokines with the four SERS nanotags that provide clearly distinguishable signals. Under Poisson distribution, the likelihood of having two or more molecules on a single pillar is <0.45% (Supplementary Table 3), which underlies the digital counting principle[24]. Compared to the theoretical Poisson distribution, the experiment data reported a close but slightly higher value, which was probably due to minor non-specific binding of SERS nanotags on the pillars.

The high sensitivity (attomolar level) of the digital nanopillar SERS assay can be ascribed to the following factors: the digital counting strategy, the single-particle SERS activity of the nanoboxes, and the use of pillars to suit confocal Raman mapping that efficiently excludes false-positive signals. Commercially available methods with potential for trace analysis of cytokines include the single-molecule ELISA Simona by Quanterix and electrochemical luminescence assay[42,43] by Meso Scale Discovery. Compared to these two methods, the developed digital nanopillar SERS platform enabled in situ multiplexed detection of four cytokines with comparable sensitivity. Unlike the issues of photo bleaching and poor multiplexing analysis often encountered in fluorescence[44] and luminescence assays[45], SERS provides the advantage of high multiplexing (e.g., 31-plex)[46–48] with the narrow Raman linewidth and high photo stability of the Raman reporters. In addition, this digital nanopillar SERS platform can provide more accurate quantification of cytokines by reducing the false-positive signals with the confocal setting, thus eventually help clinicians to monitor irAEs during immune checkpoint therapy. The highly sensitive readout for multiple targets also indicated the capability of this assay for cytokine detection to assess irAEs in clinically relevant samples.

### Evaluation of digital nanopillar SERS platform on simulated patient samples.

The detection of trace concentrations of cytokines in serum samples is difficult because plasma samples contain a high abundance of non-target molecules (e.g., serum albumin and other proteins) that can potentially interfere with cytokine detection and lead to inaccurate clinical results. To evaluate the capability of the digital nanopillar SERS assay in accurately counting single cytokine molecules, we opted to perform a recovery test in simulated patient plasma samples (i.e., healthy human serum spiked with 1 fM of FGF-2, G-CSF, GM-CSF, and CX3CL1). The rapid scan rate (i.e., 0.05 s for Raman signal integration) facilitated the detection of Raman signals from FGF-2, G-CSF, GM-CSF, and CX3CL1 SERS nanotags rather than the non-target molecules present in human serum due to their low Raman cross-section. As a representative example, Supplementary Fig. 15 shows the Raman signal distribution of the FGF-2 SERS nanotags on five different spots on the pillar array obtained from the recovery test without noticeable Raman signals from other molecules. It is worth noting that unlike the solution-based DTNB labelled SERS nanotag spectra in Fig. 1d, some of the peaks at 1556 cm$^{-1}$ and 1330 cm$^{-1}$ in Supplementary Fig. 15 had a similar intensity, which was probably because of the different orientation of the anisotropic nanoboxes on the substrate relative to the polarisation of excitation laser[49]. Supplementary Tables 4 and 5 show the cytokine concentrations in healthy

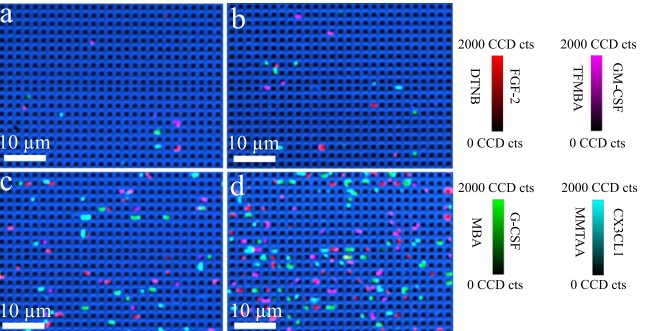

**Fig. 6 Sensitivity for the simultaneous detection of four cytokines.** Representative confocal SERS images of fibroblast growth factor 2 (FGF-2), granulocyte colony-stimulating factor (GM-CSF), granulocyte colony-stimulating factor (G-CSF), and fractalkine (CX3CL1) with the concentration of **a** 2.6 aM, **b** 26 aM, **c** 260 aM, **d** 1031 aM. Colour scale bars indicate Raman intensities from 5,5-dithiobis (2-nitrobenzoic acid) (DTNB), 4-mercaptobenzoic acid (MBA), 2,3,5,6-tetrafluoro-4-mercaptobenzoic acid (TFMBA), or 2-mercapto-4-methyl-5-thiazoleacetic acid (MMTAA). The median (interquartile range) of active pillars per scanning image of FGF-2, G-CSF, GM-CSF, CX3CL1 for 2.6 aM: 3 (1.5–3), 1 (1–2), 2 (1–3), 2 (1–3); 26 aM: 8 (5.5–10), 10 (9–13), 7 (6–10), 8 (6–10); 260 aM: 40 (36–48), 40 (35–52), 39 (35–50), 37 (36–49); and 1031 aM: 79 (61.5–97), 78 (72–87.5), 88 (68.5–97), 79 (64–95), respectively. Data represents one experiment from three independent tests.

human serum and human serum spiked with 1 fM cytokine standards determined by digital nanopillar SERS platform, respectively. On five independent pillar arrays, the measured concentrations had the relative standard deviation (RSD) below 9.0% and the Kruskal–Wallis test showed no statistical differences among these results ($p$ » 0.05). Overall, the observed inter-chip variation should enable accurate identification of disease progression to severe irAEs (e.g., grade 3 or 4), but may encounter some challenges in discriminating mild progressing to moderate irAEs (e.g., grade 1 or 2). The assay enabled trace determination of the four targets in simulated human serum as suggested by the target recovery rates of 80.00% to 137.00% with RSD from 16.02% to 21.80% (Supplementary Table 6). Importantly, the ability to measure reliably cytokines at attomolar levels in simulated human serum samples holds promise for detecting early changes in cytokine concentrations as predictors for the emergence of irAEs in immune checkpoint blockade treated patients.

To validate the accuracy of the digital nanopillar SERS assay, we compared the assay with commercially available ELISA kits (one kit for each cytokine tested) for cytokine quantification. To represent a potential clinical scenario of a patient developing irAEs during immune checkpoint blockade therapy, we prepared three samples with increasing concentrations of cytokines (spike in experiments into fetal bovine serum (FBS)) and subsequently analysed these samples with our digital nanopillar SERS assay and the commercial ELISA kits. FBS was used as complex sample matrix devoid of human cytokines. As the limits of detection for the ELISA kits (FGF-2 = 0.95 pM, G-CSF = 1.66 pM, GM-CSF = 1.11 pM, and CX3CL1 = 17.86 pM) were above the attomolar level, the simulated samples were prepared to suit the detection range of these kits. For the digital nanopillar SERS assay, the samples were diluted correspondingly and generated consistent results with the ELISA kits as shown in Supplementary Table 7. No statistical differences were found between ELISA and digital nanopillar SERS results based on Mann–Whitney test. Furthermore, we compared the detection of four cytokines in human serum with digital nanopillar assay and ELISA kits (Supplementary Table 8). The cytokine levels in human serum were below the limit of detection for the conventional ELISA kits, whereas their concentration was quantified by digital nanopillar SERS platform. For the human serum spiked with standard cytokines, the digital SERS platform generated similar results to ELISA without significant differences by Mann–Whitney test. Collectively, the digital nanopillar SERS platform showcased the ability to robustly and accurately quantify cytokines in complicated samples, which is significant for the prospect of dynamic correlation monitoring of irAEs in clinical samples.

Following the demonstration of the accuracy of digital nanopillar SERS platform, we tested the four cytokine levels in ten healthy people (Supplementary Table 9). These ten healthy people showed cytokine concentrations beyond the conventional ELISA capability to accurately quantify, which was consistent with previous reports[9,50,51].

**Dynamic correlation monitoring of irAEs in melanoma patients receiving immune checkpoint blockade treatment**. Having established the feasibility of digital nanopillar SERS in simulated clinical samples, we applied the platform for longitudinally monitoring irAEs in ten melanoma patients (2–3 time points per patient, 26 samples in total) who underwent immune checkpoint blockade therapy (Supplementary Table 10). By diluting the patient samples to follow Poisson distribution, we quantified the cytokine concentration using digital nanopillar SERS platform. Based on the clinical assessments, the patients were classified into two categories: (i) developed severe irAEs

(grades 3 and 4) and needed hospitalisation and dedicated treatment (Patients 1, 2, 3, 4, 5); and (ii) developed minor irAEs (grades 1 and 2) that could be managed with immunosuppressants (e.g., corticosteroids) or exhibited no symptom of irAEs (Patients 6, 7, 8, 9, 10).

As a representative case, Fig. 7 shows two cytokine profiles of a patient with severe irAEs (Patient 1) and a patient with mild irAEs (Patient 1). For Patient 1 who received ipilimumab (cytotoxic T-lymphocyte antigen-4 (CTLA-4) inhibitor) and was checked on days 7, 21, and 42, the confocal SERS images showed an increase in active pillars with the continuation of treatment (Fig. 7a–c), suggesting an elevation of the cytokine levels that could potentially trigger the severe irAEs. In agreement with the Raman images, the quantitative counting results for the four cytokines also corroborated the increase of cytokine concentrations peaking in sub-fM levels (Fig. 7d). These cytokine levels were below the limit of detection of conventional ELISA kits (pM level). Importantly, we observed significantly elevated cytokine concentrations in Patient 1 serum on day 42 compared to days 7 and 21. This patient showed the onset of grade 4 irAEs (i.e., colitis) 13 days later (day 55), consistent with the concept that higher cytokine levels correlate with increased risk of developing irAEs[9]. To further evaluate the utility of these four biomarkers as a signature in identifying and characterising irAEs, we analysed all the counting data from Patient 1 by applying linear discriminant analysis (LDA). As seen in Fig. 7e, the LDA successfully distinguished the data on day 42 into a separate zone from days 7 and 21, which may indicate the potential value of biomarkers in monitoring irAEs development. We further demonstrated Patient 1 LDA with the use of all combinations of two (Supplementary Fig. 16) and three cytokines (Supplementary Fig. 17). Overall, the LDA with four cytokines showed improved classification over using three or less cytokines. Interestingly, considering FGF-2/G-CSF, G-CSF/GM-CSF, and G-CSF/CX3CL1, the LDA generated similar performance to the LDA with four cytokines. To further compare the classification power of FGF-2/G-CSF, G-CSF/GM-CSF, and G-CSF/CX3CL1, and four cytokines, we performed LDA of Patient 2 (Supplementary Fig. 18), which suggested a better differentiation with the use of four cytokines. Therefore, the inclusion of all four cytokines in LDA facilitated a wider and more accurate patient sample analysis. Similarly, Patients 2, 3, 4, and 5, who manifested severe irAEs were connected with higher cytokine levels (Supplementary Fig. 19) and amelioration of irAEs symptom was witnessed with a decrease of cytokine concentrations. For these severe irAEs patients, the LDA model showed a clear discrimination in cytokine profile and this could help to identify patients at risk of irAEs (Supplementary Fig. 19).

As for Patient 6 who exhibited mild grade 2 irAEs on the skin during combined ipilimumab and pembrolizumab (programmed death-1 (PD-1) inhibitor) or single ipilimumab treatments, the dynamic monitoring displayed relatively stable cytokine levels on different follow-up visits (Fig. 7). Specifically, the confocal Raman images (Fig. 7f–h) and the molecular counting (Fig. 7i) in this patient serum consistently showed no significant cytokine level alterations on the three time points (days 0, 21, 42). Under this circumstance, LDA failed to clearly classify the data into separate sections (Fig. 7j). Likewise, Patients 7, 9, and 10 possessed stable cytokine levels and were diagnosed with low grade irAEs. Meanwhile, Patient 8 who showed decreasing cytokine levels did not display signs of irAEs. LDA was not able to classify Patients 7 and 8 who had mild irAEs and did not show irAEs, but it recognised the minor difference in Patients 9 and 10 who showed grade 1 irAEs (Supplementary Fig. 20).

Overall, we found the preliminary evidence to suggest that significantly elevated cytokine levels have a strong correlation

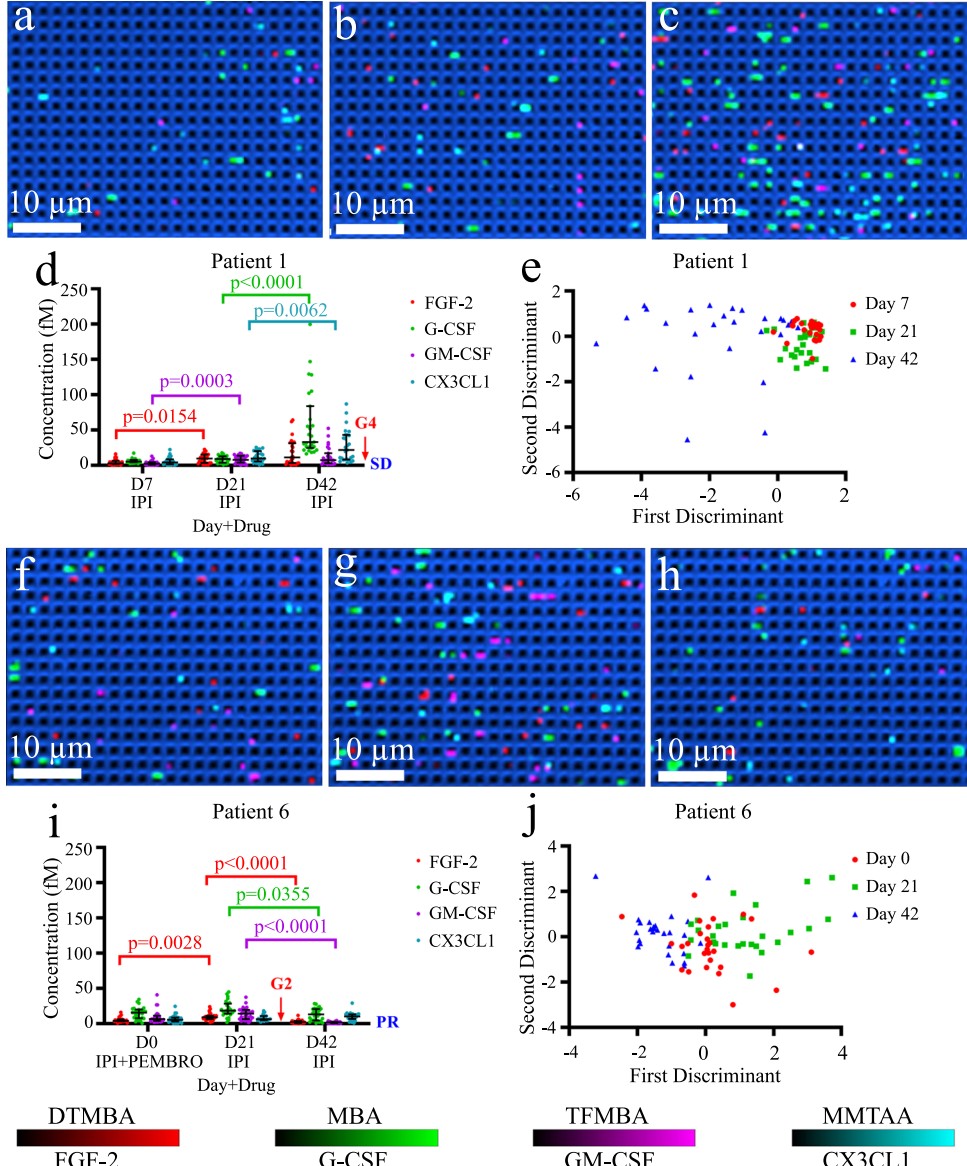

**Fig. 7 Digital nanopillar SERS assay for monitoring melanoma patients during immune checkpoint therapy.** For Patient 1 who developed severe irAEs, SERS images for cytokine detection on **a** day 7, **b** day 21, **c** day 42, **d** cytokine concentration graph for fibroblast growth factor 2 (FGF-2), granulocyte colony-stimulating factor (G-CSF), granulocyte-macrophage colony-stimulating factor (GM-CSF), and fractalkine (CX3CL1). The two shorter horizontal lines denote the interquartile ranges (25th and 75th percentile) and the longer horizontal lines in between denote the median (50th percentile), and **e** LDA analysis, respectively. For Patient 6 who developed mild irAEs, SERS images for cytokine detection on **f** day 0, **g** day 21, **h** day 42, **i** four cytokine concentration graph, the two shorter horizontal lines denote the interquartile ranges (25th and 75th percentile) and the longer horizontal lines in between denote the median (50th percentile), and **j** LDA analysis, respectively. IPI ipilimumab, PEMBRO pembrolizumab; G3 grade 3, G2 grade 2; SD stable disease, PR partial response. For Patient 1, the median (interquartile range) of active pillars per scanning image of FGF-2, G-CSF, GM-CSF, CX3CL1 on day 7: 14 (11–22.5), 23 (21, 29), 12 (7.5–18), 17 (9–25.5); day 21: 30 (19–37.5), 33 (19–41), 26 (17.5–36.5), 29 (21–43); and day 42: 33 (16.5–58.5), 76 (64–128.5), 25 (14–39.5), 48 (26.5–73.5), respectively. For Patient 6, the median (interquartile range) of active pillars per scanning image of FGF-2, G-CSF, GM-CSF, CX3CL1 on day 0: 18 (16–23), 49 (31.5–56), 23 (17.5–28), 20 (14.5–27); day 21: 29 (24–33.5), 53 (46.5–70), 35 (25–46), 22 (19–29.5); and day 42: 13 (8–16.5), 44 (23.5–55.5), 10 (6.5–12.5), 30 (24–34.5), respectively. The data represented three technical replicates obtained from three chips. Nine images were acquired from each chip for cytokine counting. Statistical analysis was based on Kruskal–Wallis test followed by Dunn's test to correct multiple comparisons (two-sided). Source data are provided in the Source Data file.

with the development and manifestation of severe irAEs, whereas stabile, low baseline, and decreasing cytokine concentration indicate mild and manageable irAEs. The relatively low concentrations of these four cytokines were below the detection sensitivities of commercially available ELISA kits (pM level), which limits their use in clinical studies. Importantly, the measurement at fM cytokine levels in clinical samples is consistent with the median concentrations of cytokine measured

by using digital ELISA[51]. The successful demonstration of the digital nanopillar SERS platform in dynamic detection of cytokines in patient serum provides a potential approach for the future accurate early detection, characterisation, and monitoring of irAEs in clinical settings. However, it is important to note that the cytokine concentration changes are not directly correlated with the treatment response according to response evaluation criteria in solid tumours (RECIST). In our pilot study,

some melanoma patients showed higher levels of cytokines compared to the healthy controls. The power of our digital nanopillar SERS platform lies in the capability to longitudinally monitor cytokines in individual patients over time.

## Discussion

Despite the frequent occurrence of irAEs in immune checkpoint therapy, particularly for the combination treatment, the prediction of the emergence of irAEs remains elusive. Mounting data suggest a potential role of cytokines as predictive markers for irAE monitoring in immune checkpoint therapy[9,11,13,52]. Although promising, accurate quantification of these biomarkers was often not possible due to the dearth in technologies with sufficient detection sensitivity. Typically, either cytokines above the detection limit of immunosorbent assay were selected[11], or relative cytokine quantification[9] was performed for investigating irAEs. The former approach has the drawback of potentially excluding the low abundance cytokines of significance in irAEs. As for relative quantification[9,53], the cytokine concentrations are determined by relating to a standard that had the observed median fluorescence value closest to the median of the test sample. The relative concentration, however, may fail to represent accurate cytokine levels and thus needs further exploration. Our developed digital nanopillar SERS assay offers early data suggestive of a potential approach to the above-mentioned challenges and provides the possibility to study trace amounts of a panel of cytokines in an accurate quantification manner as well as concomitantly providing attomolar level sensitivity. The current proof-of-principle approach has measured four of the potential inflammatory and/or immune toxicity-related cytokines for the prediction of emergence, characterisation, and/or quantifiable correlation with irAEs in melanoma patients.

By leveraging the narrow line width of Raman spectra, the developed digital SERS counting assay shows the ability to sensitively and simultaneously detect multiple cytokines. The adoption of the novel digital quantification mode in SERS using gold–silver alloy nanoboxes further improves the high sensitivity of SERS technology. Notably, the digital counting strategy offers an option for reproducible SERS quantification by avoiding the common Raman signal fluctuations induced by ensemble measurements. The ensemble measurement in SERS relies on the enhancement of Raman signals of molecules located in or near the "hot spots" (i.e., strong electromagnetic fields)[30]. Due to the random distribution and various efficiencies of "hot spots", it can result in the discrepancies in acquired SERS intensities for inter-laboratory and even intra-laboratory tests[29]. To circumvent the impact of Raman intensity fluctuation on accurate quantification, we employed the digital SERS signals from the single-SERS-active nanoboxes on discrete pillar arrays to enumerate the targets and only count the "yes" or "no" signal for a robust and reproducible SERS analysis. Furthermore, the digital readout model, which regards both aggregated and single nanoparticle as a single binding event to reflect the true target number, can have a better accuracy and robustness than the intensity-based assay[39].

We believe that the proposed digital nanopillar SERS assay could be used to monitor other cytokine-induced immune responses. For instance, with the outbreak of 2019 novel coronavirus (2019-nCoV), it is yet difficult to predict which infected patient will develop a strong immune response that requires hospitalisation. However, cytokines have been indicated to play a major role in the severity of immune response for critically ill patients infected with 2019-nCoV[54]. The specific detection of multiple cytokines at early stages of viral infection could thus potentially address this issue and help to provide the clinical care for people at the highest risk. For patients with high cytokine

concentrations, the digital nanopillar SERS platform will require the sample dilution to suit Poisson distribution.

In summary, we propose a digital nanopillar SERS platform for the parallel counting of single cytokines and dynamic monitoring in the clinical context of irAE development during immune checkpoint blockade therapy. The platform achieved attomolar level sensitivity by utilising discrete pillar array compartments to hold the single cytokine and subsequently applied single-particle active nanobox-based SERS nanotags for cytokine identification and counting. The confocal Raman mapping on the pillar array offered the highest possible clinical specificity by reducing non-specific signals and provided a "yes/no" type counting approach for reproducible Raman signal readout. The designed platform was rigorously optimised and tested in simulated clinical samples prior to the evaluation for irAE monitoring in stage IV melanoma patients receiving immune checkpoint blockade therapy. We envisaged this platform possessing the advantages of highly sensitive and multiplexing analysis capability can transit into future irAE detection methodologies after extensive validation in a large cohort of clinical samples over different time courses.

## Methods

**Materials**. Silver nitrate (AgNO₃), hydrogen tetrachloroaurate (III) trihydrate (HAuCl₄·3H₂O), MBA, DTNB, TFMBA, MMTAA, DSP were obtained from Sigma Aldrich. Ascorbic acid (AA) of analytical grade was purchased from MP Biomedicals. FGF-2 (223-FB), G-CSF (214-CS), GM-CSF (215-GM), CX3CL1 (365-FR) cytokines; monoclonal anti-FGF-2 (MAB233), anti-G-CSF (MAB214), anti-GM-CSF (MAB615), anti-CX3CL1 (MAB3652) antibodies; polyclonal anti-FGF-2 (AF-233), anti-G-CSF (AF-214), anti-GM-CSF (AF-215), anti-CX3CL1 (AF-365) antibodies; and FGF-2 (DY233-05), G-CSF (DY214-05), GM-CSF (DY215-05), and CX3CL1 (DY365) ELISA kits were bought from R&D Systems.

All the patient serum or plasma samples were collected at the Austin Hospital (Melbourne) under approved human ethic protocols and written informed consents were obtained from all patients before sample collection. Ethics approval was obtained from The University of Queensland Institutional Human Research Ethics Committee (approval nos. 2011001315 and 2016000876) and the following clinical assay was carried out according to the approved guidelines.

**Preparation of single-particle active SERS nanotags**. The preparation of SERS nanotags involved the synthesis of nanoboxes and the subsequent functionalisation with Raman reporters and antibodies. For nanobox synthesis, 45 μL of HAuCl₄ (1 wt%) was added into 10 mL of ultrapure H₂O (18.2 Ω cm) under magnetic stirring (800 r.p.m.) for 1 min, followed by simultaneously introducing 170 μL of AgNO₃ (6 mM) and 30 μL of AA (0.1 M) into the stirring solution. Then, the formation of nanoboxes was indicated by the appearance of an apparent blue colour within 6 s and the samples were collected 1 min later by centrifuging at 600 g for 15 min.

To functionalise nanoboxes with Raman reporters and antibodies, 300 μL of nanoboxes centrifuged from 1 mL of as-prepared solution were co-incubated with one type of Raman reporters (i.e., 10 μL of DTNB, 8 μL of MBA, 10 μL of TFMBA, or 10 μL of MMTAA) and 2 μL of DSP linker for 6 h. After that, the Raman reporter and DSP nanoboxes were separated by centrifuging at 600 g for 15 min, and resuspended into 300 μL of PBS (0.1 mM). Then, 2 μg of anti-FGF-2, anti-G-CSF, anti-GM-CSF, anti-CX3CL1 antibodies were added to MBA, DTNB, TFMBA, and MMTAA labelled nanoboxes, respectively. After overnight incubation at 4 ℃, the functionalised nanoboxes were purified by centrifuging at 600×g for 15 min to separate free antibodies and the final products were resuspended into 200 μL of 0.1% BSA for future use.

**Fabrication of pillar arrays**. The chip was made of a sensing array measuring 1 mm × 1 mm (Supplementary Fig. 1a) and consisted of 250,000 individual pillars. Each pillar was 1 μm wide, 1 μm long, and 1 μm high. The pillars were evenly spaced by 1 μm from one pillar to the next (Supplementary Fig. 1b). The pillar array was designed using Nanosuite 6.0 (Raith GmbH) and Beamer 5.9.1 (GenISys GmbH) and fabricated on a 4-inch p-type <100> silicon wafer (Bonda Technology Pte Ltd, Singapore) using electron beam lithography (EBL). The wafer accommodated 76 separate pillar arrays. Silicon wafer was first cleaned in acetone, iso-propanol with sonication for 2 min each, followed by rising with deionised H₂O and dehydration bake at 180 ℃ for 2 min. Prior to the resist coating, the wafer had undergone a further O₂ plasma cleaning at 200 W for 5 min (Diener Atto, Diener Electronic GmbH). The cleaned wafer was spin-coated with two layers of poly-methyl methacrylate (bottom: 495K A4 PMMA, top: 950k A4 PMMA, from MicroChemicals GmbH) using the CEE Apogee Coater (Cost Effective Equipment, LLC) at 1500 r.p.m. for 60 s each. After the coating of each layer, the wafer was baked immediately on a hot plate to prevent from intermixing of the two layers of resist. The baking time was 10 and 3 min for the bottom and top layer, respectively,

at 180 °C. The thickness of the photoresist was found to be ~450 nm (top PMMA: ~250 nm, bottom PMMA: ~200 nm), characterised by white light reflectometry (FilmTek 2000M, Scientific Computing International). EBL was performed in the Raith EBPG5150 system. The patterns were exposed in EBL with an accelerated voltage of 100 kV, 150 nA of beam current (spot size ~80 nm), with step sizes of 40 nm and an electron dose of 1200 μC/cm$^2$. The exposure time per 4-inch wafer was ~35 min, each containing 76 individual chips. After exposure, the wafer was developed in a mixture of isopropanol and methyl isobutyl ketone (3:1) for 60 s and rinsed immediately with isopropanol, followed by drying with N$_2$. An oxygen plasma descum process, at 100 W, 60 s (Diener Atto, Diener Electronic GmbH) was carried out to remove resist residues prior to the deposition. Next, 10 nm titanium and 200 nm gold were deposited by physical vapour deposition using a Temescal FC-2000 electron beam evaporator (Ferrotec, U.S.A.). After overnight lift-off at room temperature in Remover PG (MicroChemicals GmbH, Germany), the excess material was washed off and the pillar array structure was revealed (Supplementary Fig. 1c). To create the pillar height (i.e., 1 μm), reactive ion etching (Oxford Instruments, UK) was applied for anisotropic etching of the silicon. Hereby, the deposited gold served as mask to protect the underlying silicon while the unmasked silicon was removed. Next, the wafer was coated with a protective layer of cured AZnLOF 2020 prior to wafer dicing into 76 individual sensing chips consisting of a single pillar array. Prior to use, the protective layer was washed off by consecutive washes with isopropanol and acetone and dried under a stream of nitrogen.

**Pillar array functionalization.** Antibody functionalisation of the gold-topped pillar array was conducted by crosslinking the antibodies to the gold surface using DSP. A solution of 5 mM DSP in dimethyl sulfoxide was pipetted onto the pillar array and incubated at room temperature for 2 h. After rinsing the pillar array with ethanol and PBS, a solution of 5 μg/mL anti-cytokine monoclonal antibody solution (100 fold dilution of antibody stock solution) in PBS was incubated overnight at 4 °C. Subsequently, the pillar array was rinsed with PBS and blocked using 1% bovine serum albumin in PBS for 1 h. Prior to use, the pillar array was rinsed with PBS. All PBS solutions were filtered through a sterile 0.22 μm syringe filter (Millex-GP, Merck, U.S.A.).

**Digital nanopillar SERS profiling of cytokines.** Cytokines (FGF-2, G-CSF, GM-CSF, and CX3CL1) with different concentrations in PBS (2.6 aM, 26 aM, 260 aM, and 1031 aM) were incubated with antibody functionalised pillar arrays at room temperature for 30 min, followed by washing the pillar array three times with washing buffer (0.1% BSA and 0.01% Tween 20 in PBS). SERS nanotags were then added into the pillar array for another 30 min incubation under room temperature to identify the targets. Finally, the pillar arrays were washed to remove the free SERS nanotags and were subject to confocal Raman microscope for quantification. For each sample, nine SERS images with each image has the dimension of 60 μm × 48 μm were taken on the pillar array to calculate the overall cytokine concentration.

**Simulated clinical sample detection.** For the recovery experiment, standard cytokines (FGF-2, G-CSF, GM-CSF, and CX3CL1) with the concentration of 1 fM were added into healthy human serum and then diluted ten times with PBS to quantify.

For the quantification of cytokines in FBS, three simulated clinical samples were prepared by titrating various concentrations of standard cytokines into 10% FBS: Sample 1 (FGF-2 = 3.64 pM, G-CSF = 3.19 pM, GM-CSF = 4.29 pM, and CX3CL1 = 28.57 fM); Sample 2 (FGF-2 = 7.28 pM, G-CSF = 6.38 pM, GM-CSF = 8.58 pM, and CX3CL1 = 571.4 fM); and Sample 3 (FGF-2 = 14.56 pM, G-CSF = 12.76 pM, GM-CSF = 17.16 pM, and CX3CL1 = 1142.8 fM). These three samples were then detected directly using the commercial ELISA kits or digital nanopillar SERS assay with a further dilution of $10^5$, $2 \times 10^5$, and $4 \times 10^5$, respectively.

**Spectroscopic ellipsometry.** The antibody film thickness was measured by in-solution spectroscopic ellipsometry (M2000V JA Woollam Co., Inc. USA) using gold-coated substrates and flow cell (QSense® Ellipsometry, Biolin Scientific, Sweden). Measurements were performed at an angle of 65°. Data analysis was performed by CompleteEASE® software using a B-Spline data fit and Cauchy model to calculate the antibody film thickness.

**MALDI-TOF MS.** The antibody-functionalised nanopillar array chip was subjected to tryptic digest prior to analysis. Sequencing-grade trypsin was made to 50 ng/μL in 25 mM ammonium bicarbonate, and sprayed over the chip using a Bruker Imageprep instrument (Bruker, USA). After trypsin deposition, the chip was incubated in a humid environment at 40 °C for 3 h. Subsequently, the chip was sprayed with a matrix solution, 10 g/L α-cyano-4-hydroxycinnamic acid in 50% acetonitrile with 0.2% trifluoroacetic acid. Next, the chip was analysed with a Bruker Ultraflex III MALDI-TOF mass spectrometer (Bruker, USA) in positive linear mode using Flex Imaging 4.0 (Bruker, USA) with a pixel size of 60 μm. Data were collected from 2 k–30 k *m/z*, at a laser repetition rate of 200 Hz. Data were normalised using the root mean square approach and visualised using Flex Imaging 4.0 (Bruker, USA) and SCILS LAB 2017a software. For the SCILS LAB analysis, the data were imported using a convolution baseline subtraction, and displayed using root mean squared normalisation.

**Instrumentations.** SEM images of pillar arrays and nanoboxes were taken on a JEOL-7100 field emission (FE)-SEM (20 kV voltage). TEM images of nanoboxes were taken on a JEOL-2100 microscope (200 kV voltage). NTA of nanobox size distribution was performed with Malvern NanoSight NS300. UV-vis extinction spectrum of nanoboxes was performed with a Shimadzu UV-2450 spectrophotometer. Confocal Raman mapping was conducted on a WITec alpha 300 R spectrometer using 632.8 nm He–Ne laser with the power of 35 mW, a grating of 600 g/mm used with EMCCD camera, spectral resolution of 1.390 cm$^{-1}$ to 2.114 cm$^{-1}$, confocal pinhole size of 100 μm, 100× air objective with NA of 0.90, and 0.05 s integration time. The theoretical spot size was 857.80 nm based on the Abbe diffraction limit (i.e., $d = 1.22\lambda/NA$). The scanning area was set to have 60 μm × 48 μm with 86 points per line and 69 lines per image. For each pillar array, nine separate scanning areas were taken in total and the total active pillars were used for quantification. The SERS mapping images for counting were taken by focusing the laser on the top of the pillar surfaces. Specifically, the laser was firstly focused on the silicon substrates by obtaining the strongest silicon signals (520 cm$^{-1}$) and then the 100× objective was moved up in z-axis direction of 1 μm for SERS scanning. The system was calibrated with the first-order photo peak of silicon at 520 cm$^{-1}$.

**Data analysis.** To assign the SERS nanotag membership for DTNB, MBA, TFMBA, and MMTAA, Project Five 5.0 software from WITec was utilised to create four filters, which summed a spectral range of 40 cm$^{-1}$ with the centre position at the characteristic Raman peak of each reporter and subtracted the background with a polynomial algorithm. Specifically, the filter ranges of four Raman reporters DTNB-, MBA-, TFMBA-, and MMTAA-coated SERS nanotags were (1310–1350 cm$^{-1}$), (1060–1100 cm$^{-1}$), (1360–1400 cm$^{-1}$), and (1268–1308 cm$^{-1}$), respectively. All the SERS images were analysed by using threshold intensity to determine the successful binding events. Specifically, the threshold intensity of FGF-2, G-CSF, GM-CSF, and CX3CL1 was set at 5000, 4000, 5000, and 5000, respectively. For each image, the threshold intensity was doubled-checked and adjusted based on the true Raman peaks in the spectra. Statistical analysis assuming unequal variances was conducted with Kruskal–Wallis test among three groups or Mann–Whitney test between two groups with GraphPad Prism 8.4. To control the error appropriately, we performed multiple comparisons using Dunn's test. LDA of clinical samples was performed in R software (3.6.2) with the MASS package (7.3-52). The active pillars in SERS images were counted with Image J software.

## Data availability

Data supporting the findings of this work are available within this paper and the supporting information files. A reporting summary of this work is available as a Supplementary file. Source data are provided with this paper.

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

## Acknowledgements

The authors acknowledge grants received by our laboratory from the National Breast Cancer Foundation of Australia (CG-12-07) and the ARC DP (160102836 and 210103151). These grants have significantly contributed to the environment to stimulate the research described here. J.L. acknowledges support from the Australian Government Research Training Program Scholarship. A.W. and A.A.I.S. thank the National Health and Medical Research Council for funding (APP1173669 and APP1175047). A.B. is the recipient of a Fellowship from the Victorian Government Department of Health and Human Services acting through the Victorian Cancer Agency. We acknowledge the facilities, and the scientific and technical assistance, of the Australian Microscopy & Microanalysis Research Facility at the Centre for Microscopy and Microanalysis, The University of Queensland. We appreciate to receive the technical and scientific guidance from Queensland node of the Australian National Fabrication Facility (Q-ANFF) in confocal Raman mapping and spectroscopic ellipsometry measurement.

## Author contributions

J.L., A.W., A.I.S., H-H.C., Y.W., A.B., P.M., and M.T. contributed to the design of the experiments and analysing the data. J.L. and A.W. performed the experiments and prepared the manuscript. All authors read, commented, and edited the manuscript, and assisted during the revision process.

## Competing interests

The authors declare no competing interests.
