## [Peer Review File · Nature Communications]

REVIEWER COMMENTS

Reviewer #1 (Remarks to the Author):

Recommendation: Major revisions.

The author reported a digital nanopillar SERS platform which utilises a periodically patterned pillar array for confining individual cytokines, single-particle active SERS nanotags for multiplex cytokine labelling, and confocal Raman microscopy for counting individual target-specific signals. I noticed the similar concept of digital SERS platform has been proposed in *Anal. Chem.* 2018, 90, 2, 1248–1254, but I think the design of this work was meaningful for real-time multiplex single cytokine counting and enables dynamic tracking of irAEs in cancer patients receiving immune checkpoint inhibitor treatment. The manuscript seems to be interesting and neatly written with the experimental and discussion on simultaneous detection of four cytokines using the assay seems to be interesting to the clinical researchers working in this field. However, there are some important details in the experiment that the author should better think carefully before the results can be published.

My comments are listed as follows:

1. As the amount of the antibodies on each pillar was important for cytokine molecules capture and activated pillar calculated, however, the author did not discussed it at all.
2. How about the normal levels of these cytokine molecules in healthy or cancer patients before immune checkpoint blockade therapy?
3. How can you assure the discrete single cytokine on each nanopillar for the cytokines within the clinic samples setting?
4. For the dynamic correlation monitoring of irAEs in melanoma patients, the author should discuss the stability of the SERS tag with time interval.
5. How about the antibody distribution on each pillar? You should prove that all these four kinds of antibodies attached on the pillar and calculate the density.
6. I suggest that the author compared this method with the hypersensitive electrochemical luminescence MSD (Meso Scale Discovery) for the multiple factor verification. (Wendeln A C, Degenhardt K, Kaurani L, et al. Innate immune memory in the brain shapes neurological disease hallmarks[J]. *Nature*, 2018, 556(7701): 332.; Wang, Ziming, et al. "Paradoxical effects of obesity on T cell function during tumor progression and PD-1 checkpoint blockade." *Nature medicine* (2018): 1.)
7. There is insufficient result about morphology characterization and SERS property of SERS nanotags. More data should be provided.
8. The SERS mapping of nanotags on the pillar array lacks some important information such as mapping time, illumination area, spot size, laser power. Without these details, it is hard to judge the experimental results.
9. As it well known, the NIR laser irradiation (such as 785-nm incident laser) is the most commonly used for the detection of biological samples due to the sample damage issue. Many SERS-based detection method chose 785 nm excitation wavelength to reduce the damage to biological samples and the autofluorescence background. Why did you select 632 nm laser rather than 785 nm as an excitation in this study? Please discuss this situation.
10. I found the SERS signal of single nanotag was rather weak (Fig. 2c, Fig.3e-f). Such weak signal must be easily interfered or suppressed by other molecules. There are many factors in the actual biological sample would affect and interference the analytical result. So how did you deal with this problem?

Reviewer #2 (Remarks to the Author):

Review of NCOMMS-20-21281-T

Title: A novel, disruptive, single molecule digital nanopillar SERS platform for the prediction and monitoring of immune related toxicities in immune checkpoint blockade therapy

In this work, the authors present a SERS based sandwich assay to detect low concentrations of cytokines from serum samples with the ultimate goal of predicting prognosis and monitoring response of patients taking immune checkpoint inhibitors. The authors describe the fabrication of a gold-capped silicon micropillar array, functionalized with antibodies to target four specific inflammatory cytokines (FGF-2, G-CSF, GM-CSF, CSX3CL1) that have recently been associated with irEAs in melanoma patients. Using silver-gold alloy anisotropic nanoparticles conjugated with Raman reporters and detection antibodies, the authors investigate multiplex detection of the low concentration cytokines. By tuning the sample concentration and confocal Raman mapping settings, the authors claim single particle sensitivity based on the ability to count discrete particles bound in this sandwich assay. Through several control and simulated samples, the authors demonstrate the sensitivity and specificity of their system, then apply the assay to samples obtained from 10 melanoma patients undergoing different ICI therapies over several weeks. The authors associate their digital counted cytokine concentrations with changes in irAE severity and indicate potential for future applications in disease monitoring.

Novelty:

Monitoring and predicting irEAs for ICI therapy is an important and emerging field based on the widespread use of these treatments for managing cancer. Furthermore, extending the LOD to low levels of predictive cytokines could be beneficial to manage irEAs at an earlier stage and potentially mitigate issues like cessation of treatment or development of serious complications. The authors also seek to mitigate the issues of variability of SERS platforms for low level detection by utilizing a counting based approach that is less susceptible to signal intensity fluctuations.

While the work comprises a combination of methods to achieve low level detection for cytokines, there are a number of issues the authors need to address to improve clarity in order to meet that standards for publication.

The ability of the system to multiplex detection is contingent upon the optimized combination of several parameters however many significant details of the approach are omitted, making it difficult to evaluate the potential impact of this work.

The nanobox reporters are listed generating unique, narrow spectral features between 1080 and 1380 cm^{-1} which the authors claim can be used for multiplex detection. However, the authors only report the spectra of the FGF-2 targeting DTNB reporter. Furthermore, as depicted in figure 2C and figure 3E&F, the signal to noise of the reporter feature is low for the "representative" and "selected" Raman spectra indicated. There is no indication of the variation in intensity for these features, no justification for the other signal features (eg 1500-1750 cm^{-1}) that are of the same magnitude as the purportedly enhanced Raman reporter signal. Based upon the obtained signal strength and spectral resolution of the DTNB reporter, it is not clear that multiplex detection could be achieved from these Raman spectral labels.

The premise of this intensity invariant counting approach for single molecule detection is predicated on the ability to determine a binary capture and reporting event. However, the authors provide no information regarding the determination of a successful binding event, or rationale for discriminating the target specific binding from nonspecific adhesion to the substrate. The authors should report a metric (threshold intensity, peak ratio, etc) for each reporter that was reliably utilized to determine if cytokine binding occurred. This should be considered in the context of the measurement parameters that have been adopted for the study, many of which have also not been included and limit the interpretation (and therefore impact) of the work.

The authors investigate axial offset from the substrate as a means to reduce the signal from nonspecific adhesion of the cytokines or reporters to the substrate. An important concept for this determination is the NA of the objective and the confocal performance of the microscope system. The axial resolution of the system is dependent upon all of these parameters. Likewise, as the

sample is axially displaced from the confocal focus of the objective, the spot size of the beam spreads laterally and the system is more likely to collect light that is multiply scattered from other surfaces and structures. This is even more important considering that, as the authors tested by changing the grid spacing parameter of the pillar array, they are creating uniform, periodic structures that are near or below the wavelength of their laser light. This is effectively creating a reflective grating that is likely scattering light in many directions, both for the excitation light but also and SERS light that is scattered toward the substrate.

The mapping images shown throughout the manuscript indicate clusters of pixels that are counted for target binding. In figure 2, there is a large size variation in the identified SERS nanoboxes in the SEM images and the corresponding map often produces elongated features that the authors try to indicate using ellipses while the identified nanoboxes do not have this aspect ratio. The authors should provide a justification, given that this system is intended to count individual binding events, and these structures are on average 80nm (variation?) yet the mapping depicts these features across several pixels. Are these aggregated particles? If so, how do the authors account for this in their single particle counting as it will skew accuracy?

Details about reproducibility between chips for capture and counting. It seems that a single substrate was created and sliced into numerous sections for use in the study. The authors need to report the inter-array variations in counting for a single sample in order to provide an estimate of the reliability of the method for quantifying low concentrations. Directly speaks to implications of reliability for assessing patient specific samples. This is especially important given the reported RSD values as high as 30%.

The authors show controls with PBS for multiplex sensitivity however the better test would be in human (or bovine) serum. The authors should include this comparison in the recovery study, running neat/nonspiked samples to show the cytokine counts in healthy samples. Further for the simulated patient tests (and in general throughout the work), the authors should report the number of independent samples used and independent arrays imaged per condition to establish the reliability and performance of the SERS platform.

Need to supply details on the Raman system used. Laser power, grating and spectral resolution/confocal pinhole size, objective NA, immersion or air. Measurement parameters including spatial mapping step sizes/number of points/lines per scan or per pillar.

If the counting is contingent upon a sufficiently dilute cytokine sample, is there an upper limit for utilizing this with biological samples (ie cytokine storm)?

The direct comparison of the SERS array to the ELISA kits would have been more informative if the samples had been spiked with human serum as the human cytokines are of relevant interest to this study. A paired, neat serum sample and the same sample spiked with the cytokines should have been run for ELISA and then diluted for SERS array. This would provide direct pairwise comparison of the system in the context of human samples with relevant healthy variation.

The authors compare pillar size and capture efficiency using a constant number of cytokine molecules and a constant number of pillars. Given that the relative capture surface area of the pillars is much greater than the size of the cytokine or nanobox, the author hypothesize that the binding/capture efficiency is mediated by the target recognition area. Since the importance of properly tailoring the concentration of the sample to obtain Poisson distributed counting events, the authors should show that by scaling the concentration of the sample relative to target recognition surface area that the system is performing as expected. If this is not the case, an alternative explanation should be provided to justify the 1 micrometer cubic pillar structure.

In figures 4, 6, 7 and throughout the SI, the SERS maps with captured cytokine targets indicate in many areas that the signal is from a point above the silicon substrate and not above the antibody functionalized gold capped pillars. Given the sizes of the features, and the potential step sizes for data acquisition with the Witec 300R used for this study, the authors should either justify the presence of these "binding events" for counting to the non-functionalized surface, rationalize the appearance based on the measurement protocol, or use post processing to exclude data not from

the pillar capped regions (which would easily be achieved by masking the obtained map based on the Si 520cm⁻¹ peak intensity). As the authors investigated the axial offset of the collection optics to minimize nonspecific events and obtain accurate counts, these off pillar features need to be addressed with respect to minimize false positive values.

The authors dilute the concentration of individual cytokines to 1:10 pillars for Poisson criteria. For the multiplex detection, were all cytokines at 1:10 pillar concentration each or was the entire sample at 1:10 pillar concentration? Furthermore, it would be interesting for the authors to report the probability of a pillar to have 1, 2, 3, etc bound cytokine/nanobox complexes to show that this system is following the expected Poisson distribution that the authors claim they achieve, especially given the SERS labels should have spectrally distinguishable features and should be able to be resolved for each measured pixel in the SERS map.

For the melanoma samples, all the measured cytokines were fed into an LDA algorithm. However, based on the results presented in Figure 7E, only the 1st latent variable provided discrimination between stable disease and elevated cytokine levels. Were all 4 cytokine values useful for this discrimination? Did this same classification model perform well for the other grade 2 or 3 patient samples?

The authors discuss that similar cytokine quantification approaches have been used for irAEs but looking at relative values over a dynamic process. However, the authors have not provided a robust quantification of the healthy variation of these cytokines nor a threshold for determining what constitutes elevated levels, hence the information presented here is ultimately a relative metric itself. Further analysis of the recovery of individual cytokines in human serum samples is needed before the presented data can convincingly extend beyond relative values.

specific details:

Throughout the manuscript the authors provide no quantitation of the assay counting performance for tested conditions. Median and interquartile ranges should be included, at least in the captions, for all images to indicate the variability in the performance between different fields of view and chips imaged.

Melanoma patients are arbitrarily numbered and should be re-ordered into groups such for clarity (ie 1,2,3,4,5 instead of 1,2,3,7,8)

Figure 1e: digital counting box colors and counts do not match SERS map grid counts.

Figure 7d&i, figure S10, the indication of grading and disease status are difficult to read given they are located inconsistently throughout the axes. Include this information more uniformly, perhaps in the x axis.

Digital nanopillar SERS Profiling of cytokines: Nine SERS images were acquired for each sample with 60x48 micron dimensions - non overlapping? were these averaged or used as individual data points?

Instrumentation: Raman system was calibrated against silicon first-order 520 cm⁻¹ peak, not 520 nm.

Data analysis: Log transforming counting data is appropriate for parametric testing however, true Poisson distributed events have a correlation between mean and variance. therefore, the Kruskal Wallis test, assuming unequal variances is likely more appropriate for this data set of 10 patients. Furthermore, multiple comparison corrections should be made in order to control for errors appropriately. Finally, the pairwise comparisons indicated for time points in the melanoma samples are unclear; perhaps using colors to indicate comparison of interest would help.

Figure S7f, was the linear fit constrained to include the 0aM data? The data point for 2.6aM does not match the value reported in the text (log 0.35% should be approx -0.6).

Define the meaning of the error bars throughout the entire work.

Detailed Responses to the Reviewers' Comments

Reviewer #1

Remarks to the Author:

Recommendation: Major revisions.

The author reported a digital nanopillar SERS platform which utilises a periodically patterned pillar array for confining individual cytokines, single-particle active SERS nanotags for multiplex cytokine labelling, and confocal Raman microscopy for counting individual target-specific signals. I noticed the similar concept of digital SERS platform has been proposed in *Anal. Chem.* 2018, 90, 2, 1248–1254, but I think the design of this work was meaningful for real-time multiplex single cytokine counting and enables dynamic tracking of irAEs in cancer patients receiving immune checkpoint inhibitor treatment. The manuscript seems to be interesting and neatly written with the experimental and discussion on simultaneous detection of four cytokines using the assay seems to be interesting to the clinical researchers working in this field. However, there are some important details in the experiment that the author should better think carefully before the results can be published.

My comments are listed as follows:

Reply: We appreciate reviewer 1 for the valuable comments and the time to review our manuscript. As suggested by reviewer 1, the requested details in the experiments have now been provided in the revised manuscript and a point-to-point response to reviewer 1's comments is provide below.

1. As the amount of the antibodies on each pillar was important for cytokine molecules capture and activated pillar calculated, however, the author did not discussed it at all.

Reply: We agree with the reviewer that the amount of antibodies on each pillar was important.

(i) We have performed spectroscopic ellipsometry to characterise the antibody density on the pillar array. Based on the determined antibody film thickness of 18.5 nm, we then estimated the antibody density following the Cuypers model (*J. Biol. Chem.*, 1983, 258, 2426-2431): $\Gamma = d \cdot (M/A) \cdot (np^2 - 1) / (np^2 + 2)^2$, where Γ is the antibody density (mg/m²), d is the antibody film

thickness (nm), M/A is the ratio of molecular weight to molar refractivity, and n_p is refractive index of the protein. We took the values for n_p and M/A of 1.53 and 4.14 g/mL from the previous report (Appl. Surf. Sci, 2020, 518, 146269) as we used similar IgG antibody (same isotype family and similar molecular weight). The calculated antibody surface density was 5.5 mg/m², which was in agreement with the reported antibody density on substrate surfaces (Appl. Surf. Sci, 2020, 518, 146269).

(ii) Furthermore, although it is exceptionally difficult to image horizontal variation in antibody distribution on each pillar given the vast number of pillars (250,000), we attempted to image this distribution by matrix assisted laser desorption ionization-time of flight mass spectrometry (MALDI-TOF MS). It is worth stating from the outset that such techniques, because they rely on protease cleavage to obtain sufficiently volatile peptide fragments, followed by drying the cleaved films, are highly prone to drying artefacts. Nonetheless, as requested by the reviewer, we performed the MS mapping analysis. The results were very interesting. As shown in Figure S2, the antibody functionalised pillar array showed a mass spectrum with distinct high molecular weight fragments. The only source for the detected high molecular fragments was the antibodies, which suggested the successful antibody surface conjugation on the gold-topped pillars. The mapping images display the m/z signals from selected protein fragments on the chip.

Figure S2. Characterisation of antibody conjugation on the pillar array. (a) Averaged MALDI-TOF spectrum and (b) mass spectrometric mapping of pillar array that shows the distribution of antibody fragments for selected m/z of : (1) 4284, (2) 5649, (3) 6169, (4) 8482, (5) 8565, (6) 11750, (7) 12398, (8) 15497, and (9) 16979.

As suggested by the reviewer, we have now included the discussion of antibodies on pillar array in the revised manuscript on page 5 as follows.

“The successful antibody conjugation on gold-topped pillar surfaces was confirmed by matrix assisted laser desorption ionization-time of flight mass spectrometry (MALDI-TOF MS) (Figure S2), which showed high molecular weight fragments derived from antibodies. Furthermore, spectroscopic ellipsometry was utilised to estimate the antibody density on pillar surfaces. Based on the obtained film thickness of 18.5 nm, the calculated antibody surface

density was 5.5 mg/m² using the Cuypers model (J. Biol. Chem., 1983, 258, 2426-2431), which was in agreement with the reported antibody density on substrate surfaces (Appl. Surf. Sci, 2020, 518, 146269).”

2. How about the normal levels of these cytokine molecules in healthy or cancer patients before immune checkpoint blockade therapy?

Reply: Following the reviewer’s question, we have now measured the normal levels of these four cytokine molecules in 10 healthy people using the digital nanopillar SERS platform and the commercial ELISA kits. Utilising the digital nanopillar SERS platform and as shown in Table S9, the normal levels of FGF-2, G-CSF, GM-CSF, and CX3CL1 are in the range of 1.46 fM - 8.59 fM, 0.35 fM – 6.53 fM, 0.75 fM – 8.25 fM, and 1.19 fM – 7.05 fM, respectively. The fM level of cytokines in healthy people was consistent with the previous reports (Proc. Natl. Acad. Sci., 2019, 116, 4489-4495; Analyst, 2015, 140, 6277-6282). The commercial ELISA kits, however, couldn’t generate detectable signals due to the insufficient detection sensitivity.

Table S9. Concentration of 4 cytokines in the serum of 10 healthy people as determined by digital nanopillar SERS platform.

	FGF-2 (fM)	G-CSF (fM)	GM-CSF (fM)	CX3CL1 (fM)
H1	8.05±0.46	6.53±0.33	8.25±0.23	7.05±0.29
H2	5.77±1.04	1.82±0.55	3.29±0.57	3.90±0.96
H3	1.46±0.42	0.35±0.27	0.75±0.46	1.19±0.81
H4	7.40±1.36	2.15±1.09	5.87±1.09	4.48±1.38
H5	2.93±0.89	1.13±0.36	2.04±1.18	1.68±0.96
H6	2.95±0.69	1.73±0.25	2.24±0.86	1.19±0.58
H7	4.89±1.06	1.00±0.44	3.53±0.55	3.04±0.92
H8	8.59±1.05	1.57±0.67	4.69±0.40	3.16±0.32
H9	5.94±0.99	4.85±1.29	6.71±0.99	1.22±0.07
H10	1.46±0.54	1.94±0.95	1.66±0.72	1.23±0.57

The experimental data is based on three technical replicates on three chips with the data showing mean ± S.D.

For the 9 cancer patients before immune checkpoint blockade therapy, the baselines of these four cytokines were shown in the longitudinal study in Figure 7, Figure S17 and Figure S18. Specifically, the four cytokine levels of FGF-2, G-CSF, GM-CSF, and CX3CL1 were in the

range of 3.17 fM – 58.79 fM, 1.23 fM – 50.50 fM, 2.67 fM – 24.65 fM, and 3.98 fM – 82.56 fM, respectively.

We have now included the cytokine concentration of healthy human serum samples (Table S9) and the discussion into the revised manuscript on page 15 and 18 as follows.

“Following the demonstration of the accuracy of digital nanopillar SERS platform, we tested the four cytokine levels in 10 healthy people (Table S9). These 10 healthy people showed cytokine concentrations beyond the conventional ELISA capability to accurately quantify, which was consistent with previous reports (Clin. Cancer Res., 2019, 25, 1557-1563; Proc. Natl. Acad. Sci., 2019, 116, 4489-4495; Analyst, 2015, 140, 6277-6282).”

“In our pilot study, some melanoma patients showed higher levels of cytokines compared to the healthy controls. The power of our digital nanopillar SERS platform lies in the capability to longitudinally monitor cytokines in individual patients over time.”

3. How can you assure the discrete single cytokine on each nanopillar for the cytokines within the clinic samples setting?

Reply: For the clinical samples run in this work, a preliminary test of the samples was performed on the platform to ensure the discrete single cytokine on each nanopillar. This was achieved by diluting the patient samples to suit the Poisson distribution for quantification.

We have now clarified this point in the revised manuscript on page 15 as follows.

“By diluting the patient samples to follow Poisson distribution, we quantified the cytokine concentration using digital nanopillar SERS platform.”

4. For the dynamic correlation monitoring of irAEs in melanoma patients, the author should discuss the stability of the SERS tag with time interval.

Reply: Based on the reviewer’s comment, we have studied the four SERS nanotag stability by monitoring the Raman intensity over 7 days. As indicated in Figure S5, Raman intensity variations are lower than 5%, suggesting the good stability of the prepared SERS nanotags.

Figure S5. Monitoring the signal stability of the SERS nanotags over 7 days. SERS spectra of (a) FGF-2, (b) G-CSF, (c) GM-CSF, and (d) CX3CL1 nanotags on day 1, day 3, day 5, and day 7. Each spectrum represents the average measurement from 10 Raman acquisition.

As suggested by the reviewer, we have discussed the stability of the four SERS nanotags with time interval (Figure S5) in the revised manuscript on page 6 as follows.

“To investigate the SERS nanotag stability, we monitored the Raman signal intensity over 7 days. As shown in Figure S5, signal intensity variations are less than 5% in the SERS spectra, suggesting the good stability of the prepared SERS nanotags.”

5. How about the antibody distribution on each pillar? You should prove that all these four kinds of antibodies attached on the pillar and calculate the density.

Reply: We thank the reviewer for this comment. As suggested by the reviewer, we investigated the antibody distribution and calculated the density of the pillar array functionalised with four antibodies by using matrix assisted laser desorption ionization-time of flight mass spectrometry (MALDI-TOF MS) and spectroscopic ellipsometry.

To calculate the antibody density, we conducted in-solution spectroscopic ellipsometry measurement on the four antibody functionalised pillar array. Based on the determined film thickness of 18.5 nm, we then estimated the antibody density following the Cuypers model (J. Biol. Chem., 1983, 258, 2426-2431): $\Gamma = d \cdot (M/A) \cdot (n_p^2 - 1) / (n_p^2 + 2)^2$, where Γ is the antibody density (mg/m^2), d is the antibody film thickness (nm), M/A is the ratio of molecular weight to molar refractivity, and n_p is refractive index of the protein. We took the values for n_p and M/A of 1.53 and 4.14 g/mL from the previous report (Appl. Surf. Sci, 2020, 518, 146269) as we used similar IgG antibody (same isotype family and similar molecular weight). The calculated antibody surface density was $5.5 \text{ mg}/\text{m}^2$, which was in agreement with the reported antibody density on substrate surfaces (Appl. Surf. Sci, 2020, 518, 146269).

As stated in our reply to comment 1, we have attempted to image horizontally the antibody distribution on the chip by MALDI-TOF MS. Due to the similar structures of the four antibodies (all IgG type), the obtained mass spectrum can't differentiate the four antibodies based on m/z . The mass spectrum (Figure S2) shows the four antibody functionalised pillar array with distinct high molecular weight fragments. The only source for the detected high molecular fragments was the antibodies, which suggested the successful antibody surface conjugation on the gold-topped pillars. The mapping images display the m/z signals from protein fragments on the chip. It is worth stating from the outset that this technique relies on protease cleavage to obtain volatile fragments followed by drying the film and thus is prone to drying artefacts.

Figure S2. Characterisation of antibody conjugation on the pillar array. (a) Averaged MALDI-TOF spectrum and (b) mass spectrometric mapping of pillar array that shows the distribution of antibody fragments for selected m/z of : (1) 4284, (2) 5649, (3) 6169, (4) 8482, (5) 8565, (6) 11750, (7) 12398, (8) 15497, and (9) 16979.

Furthermore, our digital read-out allows digital calibration of each antibody (revised manuscript Figure 4 and 6, and SI Figure S9-12). These data clearly show, when taken into account that we have a vast redundancy of pillars, that we have the required antibody distribution to perform our digital analysis. Those data coupled with the new obtained spectroscopic ellipsometry and MALDI-TOF MS characterisations show that we have the required four antibody distribution for the digital analysis.

As suggested by the reviewer, we have added a statement about the antibody distribution and density on the pillars in the revised manuscript on page 5 as follows.

“The successful antibody conjugation on gold-topped pillar surfaces was confirmed by matrix assisted laser desorption ionization-time of flight mass spectrometry (MALDI-TOF MS) (Figure S2), which showed high molecular weight fragments derived from antibodies. Furthermore, spectroscopic ellipsometry was utilised to estimate the antibody density on pillar surfaces. Based on the obtained film thickness of 18.5 nm, the calculated antibody surface density was 5.5 mg/m² using the Cuypers model (J. Biol. Chem., 1983, 258, 2426-2431), which was in agreement with the reported antibody density on substrate surfaces (Appl. Surf. Sci., 2020, 518, 146269).”

6. I suggest that the author compared this method with the hypersensitive electrochemical luminescence MSD (Meso Scale Discovery) for the multiple factor verification. (Wendeln A C, Degenhardt K, Kaurani L, et al. Innate immune memory in the brain shapes neurological disease hallmarks [J]. Nature, 2018, 556(7701):332; Wang, Ziming, et al. "Paradoxical effects of obesity on T cell function during tumor progression and PD-1 checkpoint blockade." Nature medicine (2018): 1.)

Reply: We thank the reviewer for suggesting relevant works. Similar to our approach, the MSD assay applies a sandwich assay format where the cytokine is captured in an antibody-functionalised well. Subsequently, the captured cytokine is labelled with a ruthenium complex-conjugated antibody. After adding a substrate solution (e.g., tripropylamine) and applying an electric potential, an electrochemical luminescence signal is induced that is proportional to the cytokine concentration. The MSD assay provides the capability for spatial multiplexing. According to the website of Meso Scale Discovery (<https://www.mesoscale.com>), the assay's detection sensitivity for the four cytokines studied range from 0.08 pg/mL (4.8 fM) to 102 pg/mL (2.9 pM).

As suggested by the reviewer, we have now compared our platform with the electrochemical luminescence MSD assay and included the discussion in the revised manuscript on page 14 as follows.

“In comparison to the reported electrochemical luminescence detection (Nature, 2018, 556, 332; Nat. Med., 2019, 25, 141-151) that captured cytokines in the antibody-functionalised well,

the developed digital nanopillar SERS platform enabled in-situ multiplexed detection of four cytokines with comparable sensitivity.”

7. There is insufficient result about morphology characterization and SERS property of SERS nanotags. More data should be provided.

Reply: In the light of the reviewer’s comment, we further utilised transmission electron microscope (TEM), nanoparticle tracking analysis (NTA, NanoSight), and UV-vis extinction spectroscopy to characterise the morphological and optical property of the prepared nanoboxes (Figure S3). The TEM image suggested the hollow inner structure of the nanoboxes and the wall thickness was around 15 nm. The NTA showed the synthesised nanoboxes possessed a mode size of 77 nm. The UV-vis extinction spectroscopy demonstrated the nanoboxes had the surface plasma resonance (SPR) peak of 610 nm.

Figure S3. Nanobox morphological and optical characterisation. (a, b) TEM images of nanoboxes; (c) nanobox size distribution; and (d) UV-vis extinction spectrum of nanoboxes.

For the SERS property of nanotags, we have provided the typical four SERS nanotag spectra in Figure 1d, which suggests the nanoboxes can generate clear, characteristic, and non-overlapping Raman peaks from each Raman reporter for multiplexed detection.

Figure 1d. SERS spectra of nanoboxes conjugated with DTNB, MBA, TFMBA, or MMTAA Raman reporters.

To further evaluate the nanobox SERS property, we have calculated the enhancement factor (EF) of the four Raman reporters on the nanoboxes.

EF was calculated by the following formula:

$$EF = (I_{SERS}/N_{SERS})/(I_{RS}/N_{RS})$$

where I_{SERS} and N_{SERS} are Raman intensity and the number of molecules in SERS measurement, respectively. I_{RS} and N_{RS} are Raman intensity and the number of molecules in normal Raman measurement, respectively.

To allow a robust calculation of EF, liquid samples with Raman reporters on nanobox surfaces or dissolved in methanol were tested in a cuvette. As these four Raman reporters all utilised thiol group to attach the nanobox surfaces, we assumed they formed a self-assembly monolayer with the same coverage of 0.5 nmol/cm^2 based on the previous report (Anal. Chem., 2005, 77, 3261-3266). The nanoboxes had the concentration of 2.95×10^8 particles/mL and the mode size of 77 nm according to NTA characterisation. The concentration of Raman molecules on the nanobox surfaces was thus determined to be $0.051 \text{ }\mu\text{M}$. The normal Raman measurement was conducted with 100 mM of each Raman reporter in methanol. Figure S4 shows the representative SERS and normal Raman spectra of each Raman reporter measurement. It is worth noting that SERS spectra shifted in wavenumber compared to the normal Raman spectra

probably due to the strong interactions of molecule with gold-silver surfaces, which was similarly observed in literature (e.g., *Anal. Chem.*, 2003, 75, 5936-5943; *Molecules*, 2008, 13, 2608-2627). Based on the labelled characteristic peaks in Figure S4, the calculated EF of DTNB, MBA, TFMBA, and MMTAA was 8.14×10^6 , 1.46×10^7 , 4.01×10^7 , and 3.26×10^7 , respectively. The obtained EFs were higher than the reported spherical gold nanoparticles and pure silver nanocubes (*J. Phys. Chem. A*, 2009, 113, 3923-3939) and comparable to the reported hollow nanocubes (*J. Mater. Chem. C*, 2014, 2, 9934-9940), illustrating the high SERS property of the synthesised nanoboxes.

Figure S4. Evaluating EFs of four Raman reporters on the nanoboxes. SERS and Raman spectra of (a) DTNB, (b) MBA, (c) TFMBA, and (d) MMTAA. The concentrations of Raman reporter in SERS and normal Raman measurement was $0.051 \mu\text{M}$ and 100 mM , respectively. Each spectrum represents the average measurement from 10 Raman acquisition.

We have now provided the morphological characterisation (Figure S3), SERS spectra of each SERS nanotag in Figure 1d, EF calculation (Figure S4) in the revised manuscript on page 5-6 as follows.

“Figure S3a, 3b shows the transmission electron microscope (TEM) image of the nanoboxes with the hollow inner structure and a wall thickness of around 15 nm. Nanoparticle tracking analysis (NTA), which allows the tracking and detection of single particles, shows the

nanoboxes have a mode size of 77 nm ($D_{10}=67.6$ nm and $D_{90}=110.6$ nm) (Figure S3c). UV-vis extinction spectroscopy demonstrates the nanoboxes possess a surface plasmon resonance (SPR) peak at 610 nm (Figure S3d).”

“As shown in Figure 1d, the four SERS nanotags provide the strong and non-overlapping Raman signals, which facilitates the multiplexing analysis of four cytokines.”

“To evaluate the SERS enhancement property of the nanoboxes, we calculated the enhancement factor (EF) of the four Raman reporters on the nanoboxes. Based on the labelled characteristic peaks in Figure S4, the calculated EFs of DTNB, MBA, TFMBA, and MMTAA were 8.14×10^6 , 1.46×10^7 , 4.01×10^7 , and 3.26×10^7 , respectively. The obtained EFs were higher than the reported spherical gold nanoparticles and pure silver nanocubes (J. Phys. Chem. A, 2009, 113, 3923-3939) and comparable to the reported hollow nanocubes (J. Mater. Chem. C, 2014, 2, 9934-9940), illustrating the high SERS property of the nanoboxes.”

8. The SERS mapping of nanotags on the pillar array lacks some important information such as mapping time, illumination area, spot size, laser power. Without these details, it is hard to judge the experimental results.

Reply: We thank reviewer 1 for this comment. We agree with the reviewer and have now added the suggested SERS mapping parameters. The integration time for the mapping was 0.05 s. For each chip, we set the scanning area of $60 \mu\text{m} \times 48 \mu\text{m}$ (86 points per line and 69 lines per image) and took 9 separate images in total (i.e., illumination area of $9 \times 60 \mu\text{m} \times 48 \mu\text{m}$). The theoretical spot size was 857.80 nm based on the Abbe diffraction limit (i.e., $d=1.22\lambda/\text{NA}$, in which $\lambda=632.8$ nm, $\text{NA}=0.9$). The laser has the power of 35 mW.

We have now provided these details to the experimental section of the revised manuscript on page 23.

“Confocal Raman mapping was conducted on a WITec alpha 300 R spectrometer using 632.8 nm He-Ne laser with the power of 35 mW, a grating of 600 g/mm used with EMCCD camera, spectral resolution of 1.390 cm^{-1} to 2.114 cm^{-1} , confocal pinhole size of $100 \mu\text{m}$, $100\times$ air objective with NA of 0.90, and 0.05 s integration time. The theoretical spot size was 857.80 nm based on the Abbe diffraction limit (i.e., $d=1.22\lambda/\text{NA}$). The scanning area was set to have

60 $\mu\text{m} \times 48 \mu\text{m}$ with 86 points per line and 69 lines per image. For each pillar array, nine separate scanning areas were taken in total and the total active pillars were used for quantification.”

9. As it well known, the NIR laser irradiation (such as 785-nm incident laser) is the most commonly used for the detection of biological samples due to the sample damage issue. Many SERS-based detection method chose 785 nm excitation wavelength to reduce the damage to biological samples and the autofluorescence background. Why did you select 632 nm laser rather than 785 nm as an excitation in this study? Please discuss this situation.

Reply: We thank reviewer 1 for this insightful comment. The nanoboxes had the surface plasmon resonance (SPR) peak around 610 nm as shown in the UV-vis extinction spectroscopy in Figure S3d. Compared to 785 nm excitation, 632.8 nm laser is more close to the SPR peak (610 nm) of the nanoboxes and thus has a stronger Raman signal enhancement effect (Chem. Rev., 2013, 113, 1391-1428). Furthermore, 632.8 nm laser has a higher Raman scattering efficiency than 785 nm laser as the scattering efficiency is inversely proportional to λ^4 (λ =laser wavelength). Therefore, 632.8 nm laser was utilised as the excitation source for a more sensitive signal readout.

Figure S3d. UV-vis extinction spectrum of nanoboxes.

As suggested by reviewer 1, we have now included the discussion on the selection of laser wavelength (632.8 nm) in the revised manuscript on page 6 as follows.

“UV-vis extinction spectroscopy demonstrates the nanoboxes possess a surface plasmon resonance (SPR) peak at 610 nm (Figure S3d). The resonance frequency of the nanoboxes enables a more sensitive signal readout with 632.8 nm laser excitation (Chem. Rev., 2013, 113, 1391-1428), which also has a higher Raman scattering efficiency than 785 nm laser.”

10. I found the SERS signal of single nanotag was rather weak (Fig. 2c, Fig.3e-f). Such weak signal must be easily interfered or suppressed by other molecules. There are many factors in the actual biological sample would affect and interference the analytical result. So how did you deal with this problem?

Reply: To excite the normal Raman signals of the biological molecules with low Raman cross-section, a much longer integration time is typically needed (e.g., 15 s in Chem. Sci., 2020, 11, 525-533; 30 s in Vib. Spectrosc., 2020, 109, 103073). In our experiments, we employed 0.05 s integration time per point for a fast SERS scan and this short interrogation reduced the potential interference from other biological molecules.

We further showed five representative SERS spectra of FGF-2 SERS nanotags in detecting FGF-2 in human serum (Figure S15). As it clearly indicates in Figure S15, the peaks at 1330 cm^{-1} and 1556 cm^{-1} from DTNB reporter in the SERS nanotags are distinct and no signal from the interferences was observed in human serum.

Figure S15. Representative SERS spectra of FGF-2 SERS nanotags in detecting FGF-2 in human serum. The two labelled peaks are from DTNB reporters in FGF-2 SERS nanotags.

We have now included this discussion (Figure S15) in the revised manuscript on page 14 as follows.

“Figure S15 shows five representative SERS spectra on the pillars for the detection of FGF-2 in human serum. Due to the rapid scan (0.05 s Raman integration), only Raman signals on FGF-2 SERS nanotags were observed without interference from other molecules in the human serum sample.”

Reviewer #2

(Remarks to the Author):

Review of NCOMMS-20-21281-T

Title: A novel, disruptive, single molecule digital nanopillar SERS platform for the prediction and monitoring of immune related toxicities in immune checkpoint blockade therapy

In this work, the authors present a SERS based sandwich assay to detect low concentrations of cytokines from serum samples with the ultimate goal of predicting prognosis and monitoring response of patients taking immune checkpoint inhibitors. The authors describe the fabrication of a gold-capped silicon micropillar array, functionalized with antibodies to target four specific inflammatory cytokines (FGF-2, G-CSF, GM-CSF, CX3CL1) that have recently been associated with irAEs in melanoma patients. Using silver-gold alloy anisotropic nanoparticles conjugated with Raman reporters and detection antibodies, the authors investigate multiplex detection of the low concentration cytokines. By tuning the sample concentration and confocal Raman mapping settings, the authors claim single particle sensitivity based on the ability to count discrete particles bound in this sandwich assay. Through several control and simulated samples, the authors demonstrate the sensitivity and specificity of their system, then apply the assay to samples obtained from 10 melanoma patients undergoing different ICI therapies over several weeks. The authors associate their digital counted cytokine concentrations with changes in irAE severity and indicate potential for future applications in disease monitoring.

Novelty:

Monitoring and predicting irEAs for ICI therapy is an important and emerging field based on the widespread use of these treatments for managing cancer. Furthermore, extending the LOD

to low levels of predictive cytokines could be beneficial to manage irAEs at an earlier stage and potentially mitigate issues like cessation of treatment or development of serious complications. The authors also seek to mitigate the issues of variability of SERS platforms for low level detection by utilizing a counting based approach that is less susceptible to signal intensity fluctuations.

While the work comprises a combination of methods to achieve low level detection for cytokines, there are a number of issues the authors need to address to improve clarity in order to meet that standards for publication.

The ability of the system to multiplex detection is contingent upon the optimized combination of several parameters however many significant details of the approach are omitted, making it difficult to evaluate the potential impact of this work.

Reply: We are grateful to the reviewer for the positive evaluation of our manuscript and thoughtful comments. We have addressed all the comments made by reviewer 2 and revised the manuscript in the light of these comments. We have now provided the suggested details of the approach, and further improved the discussion and clarity of the revised manuscript.

The nanobox reporters are listed generating unique, narrow spectral features between 1080 and 1380 cm^{-1} which the authors claim can be used for multiplex detection. However, the authors only report the spectra of the FGF-2 targeting DTNB reporter. Furthermore, as depicted in figure 2C and figure 3E&F, the signal to noise of the reporter feature is low for the "representative" and "selected" Raman spectra indicated. There is no indication of the variation in intensity for these features, no justification for the other signal features (eg 1500-1750 cm^{-1}) that are of the same magnitude as the purportedly enhanced Raman reporter signal. Based upon the obtained signal strength and spectral resolution of the DTNB reporter, it is not clear that multiplex detection could be achieved from these Raman spectral labels.

Reply: We thank reviewer 2 for this excellent comment. The low signal to noise of reporter feature was due to the short integration time (0.05 s per point). As the quantification was based on counting instead of intensity, we employed this short integration time to allow a fast scan as well as identifiable signals for digital readout. To demonstrate the signal variation, we further showed the SERS spectra from individual nanoboxes (n=9) in Figure S6, which had the

average and standard deviation of SERS signal intensity of 213.41 and 85.03, respectively. As shown in Figure S6, the individual nanobox Raman signals didn't show significant variations to influence their identification and counting.

Figure S6. DTNB Raman spectra from individual nanoboxes (n=9).

In terms of the justification of signal features (e.g., 1500-1750 cm^{-1}) generated from the DTNB labelled SERS nanotags, we have included the assignment for the vibrational mode of the four SERS nanotags in the revised manuscript in Table S1. For instance, the two clearly enhanced peaks in DTNB (i.e., 1330 cm^{-1} and 1566 cm^{-1}) from Figure 2 and 3 can be assigned to the symmetric nitro stretching and the aromatic ring vibration, respectively (Anal. Chem., 2003, 75, 5936-5943).

Table S1. Assignment of SERS peaks from four Raman reporters enhanced by nanoboxes.

Raman reporter	Raman peak (cm^{-1})	Assignment
DTNB	1060	Succinimidyl N-C-O stretching and aromatic ring vibration ¹
DTNB	1330	Symmetric NO_2 stretching ¹
DTNB	1556	Aromatic ring vibration ¹
MBA	1080	Aromatic ring vibration ²
MBA	1580	Aromatic ring vibration ²
TFMBA	1380	CH_2 deformation ³
TFMBA	1631	NH_2 deformation ³
MMTAA	1288	CH in-plane bending ⁴

References:

1. Anal. Chem., 2003, 75, 5936-5943

2. J. Colloid Interface Sci., 2015, 438, 116-121
3. J. Mol. Struct., 2018, 1159, 103-117
4. Theranostics, 2018, 8, 941-954

To clearly display the Raman spectral peaks from the four Raman labels, we have integrated the SERS spectra of individual SERS nanotags into Figure 1d, which covers the spectral range from 1000-1800 cm^{-1} . As illustrated in Figure 1d, these four Raman reporters generate non-overlapping characteristic Raman signals, which enables the multiplex cytokine counting.

Figure 1d. SERS spectra of nanoboxes conjugated with DTNB, MBA, TFMBA, or MMTAA Raman reporters.

We have now included the individual nanobox SERS spectra (Figure S6), the assignment of Raman peaks from the four Raman reporters, and the discussion of four Raman reporter SERS signals (Figure 1d) to the revised manuscript on pages 6-7, as follows.

“Figure S6 further shows the SERS spectra from other individual nanoboxes. All the spectra displayed identifiable peaks for counting with an average intensity of 213.41 and standard deviation of 85.03, respectively.”

“The assignment of the major Raman peaks from the four Raman reporters were summarised into Table S1.”

“As shown in Figure 1d, the four SERS nanotags provide the strong and non-overlapping Raman signals, which facilitates the multiplexing analysis of four cytokines.”

The premise of this intensity invariant counting approach for single molecule detection is predicated on the ability to determine a binary capture and reporting event. However, the authors provide no information regarding the determination of a successful binding event, or rationale for discriminating the target specific binding from nonspecific adhesion to the substrate. The authors should report a metric (threshold intensity, peak ratio, etc) for each reporter that was reliably utilized to determine if cytokine binding occurred. This should be considered in the context of the measurement parameters that have been adopted for the study, many of which have also not been included and limit the interpretation (and therefore impact) of the work.

Reply: To clarify, we used a threshold intensity to determine the binding of SERS nanotags on pillar array. The threshold intensity for FGF-2, G-CSF, GM-CSF, and CX3CL1 was set at 5000, 4000, 5000, and 5000, respectively. The selection of different threshold values for the different SERS nanotags was to account for the differences in Raman signal intensity of the Raman reporter. For each image, the threshold intensity was doubled-checked and adjusted based on the true Raman peaks in the spectra.

As suggested by reviewer 2, we have now provided the threshold intensity and the description in the data analysis on page 23 as follows.

“The confocal Raman images were analysed with the WITec Project FIVE 5.0. All the SERS images were analysed by using threshold intensity to determine the successful binding events. Specifically, the threshold intensity of FGF-2, G-CSF, GM-CSF, and CX3CL1 was set at 5000, 4000, 5000, and 5000, respectively. For each image, the threshold intensity was doubled-checked and adjusted based on the true Raman peaks in the spectra.”

The authors investigate axial offset from the substrate as a means to reduce the signal from nonspecific adhesion of the cytokines or reporters to the substrate. An important concept for this determination is the NA of the objective and the confocal performance of the microscope

system. The axial resolution of the system is dependent upon all of these parameters. Likewise, as the sample is axially displaced from the confocal focus of the objective, the spot size of the beam spreads laterally and the systems is more likely to collect light that is multiply scattered from other surfaces and structures. This is even more important considering that, as the authors tested by changing the grid spacing parameter of the pillar array, they are creating uniform, periodic structures that are near or below the wavelength of their laser light. This is effectively creating a reflective grating that is likely scattering light in many directions, both for the excitation light but also and SERS light that is scattered toward the substrate.

Reply: We thank reviewer 2 for these thoughtful comments. In our Raman mapping measurements, we used 632.8 nm laser and 100×objective with the NA of 0.90. The theoretical lateral resolution should be 428.90 nm based on Rayleigh criterion ($0.61\lambda/NA$). Considering the complexity of Raman scattering, the typical Raman lateral resolution is higher than the theoretical value and is in the order of 1 μm (Anal. Chem., 2010, 82, 2608-2611). The designed pillar array spacing was thus controlled at 1 μm to allow the clear differentiation of adjacent pillars for obtaining the periodic structures.

To reduce the signals from nonspecific adhesion on silicon substrate, we controlled the position of the 100 \times objective by using the “Microscope Z control” instead of the sample “Scan Table”. Specifically, we firstly focused the laser on the silicon substrates by obtaining the strongest silicon signals (520 cm^{-1}) and then the 100 \times objective was moved up in z-axis direction for 1 μm to make the laser focus on the gold-topped pillar surfaces for SERS scanning. By enabling the laser to directly focus on the pillar surfaces, we avoided the spread of laser beam during the mapping.

As suggested by reviewer 2, we have now clarified this point in the optimisation section on page 8 and included the experimental details in our revised manuscript on page 23 as follows.

“The Raman mappings were performed by moving the objective along the z-axis direction with different heights (0 nm, 500 nm, 1000 nm, and 1500 nm) to compare the signal intensity.”

“The SERS mapping images for counting were taken by focusing the laser on the top of the pillar surfaces. Specifically, the laser was firstly focused on the silicon substrates by obtaining

the strongest silicon signals (520 cm^{-1}) and then the $100\times$ objective was moved up in z-axis direction of $1\text{ }\mu\text{m}$ for SERS scanning.”

The mapping images shown throughout the manuscript indicate clusters of pixels that are counted for target binding. In figure 2, there is a large size variation in the identified SERS nanoboxes in the SEM images and the corresponding map often produces elongated features that the authors try to indicate using ellipses while the identified nanoboxes do not have this aspect ratio. The authors should provide a justification, given that this system is intended to count individual binding events, and these structures are on average 80nm (variation?) yet the mapping depicts these features across several pixels. Are these aggregated particles? If so, how do the authors account for this in their single particle counting as it will skew accuracy?

Reply: The reason for the discrepancy between nanobox size and SERS mapping signals (shape of the signals) was because of insufficient mapping resolution. Because of the intricacies of Raman particle imaging, especially for the irregular sized particles, the optical resolution of the system is around $1\text{ }\mu\text{m}$ (Anal. Chem., 2010, 82, 2608-2611). As the average size of nanoboxes was much lower than Raman mapping resolution, the Raman mapping image couldn't truly reflect the geometric features of the nanoboxes.

To further characterise the ensemble nanobox size distribution, we conducted nanoparticle tracking analysis (NTA) measurements. As shown in Figure S3c, the synthesised nanoboxes possessed a mode size of 77 nm ($D_{10}=67.6\text{ nm}$ and $D_{90}=110.6\text{ nm}$).

Figure S3c. Nanobox size distribution determined by NTA.

As the digital signal readout mode was based on counting instead of signal intensity, the aggregated and single-particle were unanimously regarded as a single binding event that reflects the true target number, which means the result accuracy will not be skewed (Anal. Chem., 2019, 91, 9435-9441).

In the light of reviewer 2's comment, we have now clarified the Raman mapping images, provided the nanobox size distribution (Figure S3c), and included the discussion in the revised manuscript on pages 7, 6, and 19, respectively as follows.

“Because of the intricacies of Raman particle imaging, especially for the irregular sized particles, the optical resolution of the system is around 1 μm (Anal. Chem., 2010, 82, 2608-2611). The elongated bright Raman spots, which couldn't truly reflect the geometric features of the nanoboxes, were due to the fact that the average size of nanoboxes was much lower than Raman mapping resolution.”

“Nanoparticle tracking analysis (NTA), which allows the tracking and detection of single particles, shows the nanoboxes have a mode size of 77 nm (D10=67.6 nm and D90=110.6 nm) (Figure S3c).”

“Furthermore, the digital readout model, which regards both aggregated and single nanoparticle as a single binding event to reflect the true target number, can have a better accuracy and robustness than the intensity based assay (Anal. Chem., 2019, 91, 9435-9441).”

Details about reproducibility between chips for capture and counting. It seems that a single substrate was created and sliced into numerous sections for use in the study. The authors need to report the inter-array variations in counting for a single sample in order to provide an estimate of the reliability of the method for quantifying low concentrations. Directly speaks to implications of reliability for assessing patient specific samples. This is especially important given the reported RSD values as high as 30%.

Reply: As suggested by reviewer 2, we performed additional experiments to determine the inter-array variations. We tested the blank human serum and serum spiked with 1 fM four cytokines on five separate nanopillar chips. As shown in Table S4 and Table S5, the RSDs

among five chip measurements were below 8.2% and the p values were higher than 0.05 by Kruskal Wallis test, illustrating no significant inter-array variations.

Table S4. Detection of cytokines in healthy (blank) human serum on five independent chips.

Serum	Chip 1 (fM)	Chip 2 (fM)	Chip 3 (fM)	Chip 4 (fM)	Chip 5 (fM)	RSD (%)	p Value ^a
FGF-2	8.29	8.45	7.40	7.10	7.58	6.72	0.49
G-CSF	6.19	6.98	6.42	6.14	7.15	6.28	0.56
GM-CSF	8.01	8.57	8.16	7.66	7.94	3.69	0.95
CX3CL1	7.28	6.63	7.22	5.87	6.27	8.18	0.22

^aKruskal Wallis test. A nonparametric test to statistically determine the significant differences between two or more groups.

Table S5. Detection of cytokines in human serum spiked with 1 fM standards on five independent chips.

Spiked Serum	Chip 1 (fM)	Chip 2 (fM)	Chip 3 (fM)	Chip 4 (fM)	Chip 5 (fM)	RSD (%)	p Value ^a
FGF-2	9.11	9.90	8.34	8.55	9.11	5.52	0.88
G-CSF	7.40	8.00	8.02	7.47	8.82	5.86	0.84
GM-CSF	8.81	9.45	9.00	8.79	9.22	2.54	0.95
CX3CL1	8.06	7.55	7.99	6.44	7.24	7.20	0.68

^aKruskal Wallis test. A nonparametric test to statistically determine the significant differences between two or more groups.

Based on the additional new data, we further calculated the recovery rate and summarised the data in Table S6.

Table S6. Recovery test of four cytokines in spiked human serum.

Cytokines	Added (fM)	Detected (fM)	Recovery (%)	RSD (%)
FGF-2	1.00	1.24	124.00	21.80
G-CSF	1.00	1.37	137.00	16.19
GM-CSF	1.00	0.99	99.00	17.29
CX3CL1	1.00	0.80	80.00	16.02

As suggested by reviewer 2, we have now included the discussion of inter-array variation (Table S4 and S5) and the updated the recovery (Table S6) in the revised manuscript on page 14.

“Table S3 and S4 show the cytokine concentrations in healthy human serum and human serum spiked with 1 fM cytokine standards determined by digital nanopillar SERS platform, respectively. On five independent pillar arrays, the measured concentrations had the relative standard deviation (RSD) below 9.0% and the Kruskal Wallis test showed no statistical differences among these results ($p \gg 0.05$). The assay enabled trace determination of the four targets in simulated human serum as suggested by the target recovery rates of 80.00% to 137.00% with RSD from 16.02% to 21.80% (Table S6).”

The authors show controls with PBS for multiplex sensitivity however the better test would be in human (or bovine) serum. The authors should include this comparison in the recovery study, running neat/nonspiked samples to show the cytokine counts in healthy samples. Further for the simulated patient tests (and in general throughout the work), the authors should report the number of independent samples used and independent arrays imaged per condition to establish the reliability and performance of the SERS platform.

Reply: As suggested by the reviewer, we conducted the multiplex sensitivity in human serum (Figure S14). Because of the more complicated sample matrix composition, the lowest detectable cytokine concentration (5.2 aM) was higher than the PBS solution (2.6 aM).

Figure S14. Sensitivity for the simultaneous detection of four cytokine molecules in human serum. Linear relationship curve for the detection of (a) FGF-2, (b) G-CSF, (c) GM-CSF, and (d) CX3CL1, respectively. The error bars represented the standard deviation from three independent technical measurements on three chips.

As for the recovery study, we have now provided the cytokine concentration in healthy samples in Table S4.

Table S4. Detection of cytokines in healthy (blank) human serum on five independent chips.

Serum	Chip 1 (fM)	Chip 2 (fM)	Chip 3 (fM)	Chip 4 (fM)	Chip 5 (fM)	RSD (%)	p Value ^a
FGF-2	8.29	8.45	7.40	7.10	7.58	6.72	0.49
G-CSF	6.19	6.98	6.42	6.14	7.15	6.28	0.56
GM-CSF	8.01	8.57	8.16	7.66	7.94	3.69	0.95
CX3CL1	7.28	6.63	7.22	5.87	6.27	8.18	0.22

^aKruskal Wallis test. A nonparametric test to statistically determine the significant differences between two or more groups.

For the simulated patient tests, each sample was measured with three technical replicates on three independent chips in digital nanopillar SERS platform or three independent wells in ELISA assay. We have now run the Mann-Whitney test to compare the digital nanopillar SERS platform and ELISA measurements. For all the simulated patient samples, the p values between these two assays were higher than 0.05, which suggested the reliability of the digital SERS platform.

According to the reviewer's suggestions, we have now provided the multiplex sensitivity in human serum (Figure S14), the cytokine concentration in healthy samples (Table S4), and reported simulated patient test on pages 13-15 as follows.

“To further investigate the multiplexing quantification performance of the digital nanopillar SERS assay in human serum, we spiked standard cytokines in human serum and tested the dynamic range. Figure S14 shows the linear relationship curves for the four targets. Because of the more complicated sample matrix composition in human samples, the lowest detectable cytokine concentration (5.2 aM) was higher than the PBS solution (2.6 aM).”

“Table S3 and S4 show the cytokine concentrations in healthy human serum and human serum spiked with 1 fM cytokine standards determined by digital nanopillar SERS platform, respectively.”

“No statistical differences were found between ELISA and digital nanopillar SERS results based on Mann-Whitney test.”

Need to supply details on the Raman system used. Laser power, grating and spectral resolution/confocal pinhole size, objective NA, immersion or air. Measurement parameters including spatial mapping step sizes/number of points/lines per scan or per pillar.

Reply: We used the Raman microscope that is equipped with 632.8 nm laser with the power of 35 mW, a grating of 600 g/mm used with EMCCD camera, spectral resolution of 1.390 cm⁻¹ to 2.114 cm⁻¹, confocal pinhole size of 100 μm, and 100 × air objective with NA of 0.90. For our Raman mapping, we set each image to have 60 μm×48 μm with 86 points per line and 69 lines per image.

We agree with reviewer 2's comment and have now provided all these details in the experimental section in the revised manuscript on page 23 as follows.

“Confocal Raman mapping was conducted on a WITec alpha 300 R spectrometer using 632.8 nm He-Ne laser with the power of 35 mW, a grating of 600 g/mm used with EMCCD camera, spectral resolution of 1.390 cm^{-1} to 2.114 cm^{-1} , confocal pinhole size of 100 μm , 100 \times air objective with NA of 0.90, and 0.05 s integration time. The theoretical spot size was 857.80 nm based on the Abbe diffraction limit (i.e., $d=1.22\lambda/\text{NA}$). The scanning area was set to have 60 $\mu\text{m}\times 48 \mu\text{m}$ with 86 points per line and 69 lines per image. For each pillar array, nine separate scanning areas were taken in total and the total active pillars were used for quantification.”

If the counting is contingent upon a sufficiently dilute cytokine sample, is there an upper limit for utilizing this with biological samples (ie cytokine storm)?

Reply: There is no upper limit *per se*. For the biological samples with extremely high cytokine concentrations, sample dilutions is required to suit the Poisson distribution for quantification.

We have clarified this point in the discussion section in the revised manuscript on page 19 as follows.

“For patients with extremely high cytokine concentrations, the digital nanopillar SERS platform will require the sample dilution to suit Poisson distribution.”

The direct comparison of the SERS array to the ELISA kits would have been more informative if the samples had been spiked with human serum as the human cytokines are of relevant interest to this study. A paired, neat serum sample and the same sample spiked with the cytokines should have been run for ELISA and then diluted for SERS array. This would provide direct pairwise comparison of the system in the context of human samples with relevant healthy variation.

Reply: Based on the reviewer's comment, we have now tested a paired serum samples using digital nanopillar SERS platform and ELISA (Table S8). In neat human serum, the cytokine levels were below the limit of detection of the conventional ELISA test, whereas their concentration was quantified by digital nanopillar SERS platform. For the human serum spiked

with standard cytokines, the digital SERS platform generated similar results to the ELISA without significant differences by Mann-Whitney test (Table S8).

Table S8. Detection of cytokines in human serum and spiked serum samples with digital nanopillar SERS platform and ELISA assay. The spiked FGF-2, G-CSF, GM-CSF, and CX3CL1 concentrations in human serum were 14.56 pM, 12.76 pM, 17.16 pM, and 114.28 pM, respectively.

Cytokine	Serum		Spiked serum	
	SERS (fM)	ELISA	SERS (pM)	ELISA (pM)
FGF-2	8.05±0.46	N/A	19.49±1.51	20.20±0.50
G-CSF	6.53±0.33	N/A	11.26±2.12	10.77±0.53
GM-CSF	8.25±0.23	N/A	16.06±1.88	18.76±0.06
CX3CL1	7.05±0.29	N/A	169.22±40.77	171.02±16.58

N/A: not applicable

We have now provided the comparison of digital nanopillar SERS platform and ELISA assay in human serum (Table S8) in the revised manuscript on page 15 as follows.

“Furthermore, we compared the detection of four cytokines in human serum with digital nanopillar assay and ELISA kits (Table S8). The cytokine levels in human serum were below the limit of detection for the conventional ELISA kits, whereas their concentration was quantified by digital nanopillar SERS platform. For the human serum spiked with standard cytokines, the digital SERS platform generated similar results to ELISA without significant differences by Mann-Whitney test.”

The authors compare pillar size and capture efficiency using a constant number of cytokine molecules and a constant number of pillars. Given that the relative capture surface area of the pillars is much greater than the size of the cytokine or nanobox, the author hypothesize that the binding/capture efficiency is mediated by the target recognition area. Since the importance of properly tailoring the concentration of the sample to obtain Poisson distributed counting events, the authors should show that by scaling the concentration of the sample relative to target

recognition surface area that the system is performing as expected. If this is not the case, an alternative explanation should be provided to justify the 1 micrometer cubic pillar structure.

Reply: We thank reviewer 2 for this excellent comment. As the accessible target recognition surface area per pillar increases, it can possibly promote the thermodynamics and kinetics for higher surface binding and capture efficiency (Nat. Biomed. Eng., 2019, 3, 438-451). Following the reviewer's comment, we further tested FGF-2 sample at the concentration of 260 aM (i.e., theoretical active pillar percentage of 2.5%) on pillar array chips with three different sizes and compared with the active pillar obtained at a higher concentration of 1031 aM (i.e., theoretical active pillar percentage of 10.00%). As shown in Table S2, the active pillar percentage at both concentrations increases with the pillar size increasing, which agrees with target recognition area per pillar.

Table S2. Optimisation of target capture efficiency using pillar array with three sizes.

	250 nm active pillar%		500 nm active pillar%		1000 nm active pillar%	
	Theoretical	Experimental	Theoretical	Experimental	Theoretical	Experimental
1031aM	10.00	1.90	10.00	5.90	10.00	10.05
260 aM	2.50	0.37	2.50	1.00	2.50	2.08

We have now clarified this aspect in the revised manuscript on page 10 as follows.

“We further tested a sample with 260 aM FGF-2 (i.e., 2.5% active pillars) on the pillar array chips with 250, 500, and 1000 nm pillar widths. The capture efficiency of these three chips was summarised in Table S2. In comparison to the pillar array of 250 nm and 500 nm sizes, the 1000 nm provided an improved capture efficiency. As the accessible target recognition surface area per pillar increases, it can possibly promote the thermodynamics and kinetics for higher surface binding and capture efficiency (Nat. Biomed. Eng., 2019, 3, 438-451).”

In figures 4, 6, 7 and throughout the SI, the SERS maps with captured cytokine targets indicate in many areas that the signal is from a point above the silicon substrate and not above the antibody functionalized gold capped pillars. Given the sizes of the features, and the potential step sizes for data acquisition with the Witec 300R used for this study, the authors should either justify the presence of these "binding events" for counting to the non-functionalized surface, rationalize the appearance based on the measurement protocol, or use post processing to exclude data not from the pillar capped regions (which would easily be achieved by masking

the obtained map based on the Si 520cm⁻¹ peak intensity). As the authors investigated the axial offset of the collection optics to minimize nonspecific events and obtain accurate counts, these off pillar features need to be addressed with respect to minimize false positive values.

Reply: We thank the reviewer for the suggestion. We have now removed the signals on silicon substrate and updated the data in the revised manuscript.

The authors dilute the concentration of individual cytokines to 1:10 pillars for Poisson criteria. For the multiplex detection, were all cytokines at 1:10 pillar concentration each or was the entire sample at 1:10 pillar concentration? Furthermore, it would be interesting for the authors to report the probability of a pillar to have 1, 2, 3, etc bound cytokine/nanobox complexes to show that this system is following the expected Poisson distribution that the authors claim they achieve, especially given the SERS labels should have spectrally distinguishable features and should be able to be resolved for each measured pixel in the SERS map.

Reply: In the multiplex experiment, we controlled each target at 1:10 ratio as these four cytokines independently followed Poisson distribution.

To study the experimental Poisson distribution, we used the chips with four cytokines and SERS nanotags to have a total of 10% active pillars (i.e., each cytokine activated 2.5% pillars). The experimental probability of finding 1, 2, 3, and 4 molecules on a pillar was calculated by counting 1, 2, 3, and 4 types of specific Raman reporter peaks on a pillar in the SERS mapping images. The theoretical probability was determined with the formula $P(X=k) = \frac{\lambda^k e^{-\lambda}}{k!}$, where k is the number of molecules observed on a single pillar (i.e., 1, 2, 3, 4), λ is expected molecule number on a single pillar (i.e., 0.1), e is Euler's number. We summarised the experimental and theoretical probability in Table S1. Generally, the probability of having more than one cytokine molecule per pillar is very low by controlling the active pillars within 10%. The experimental probability correlated with theoretical value but had a slightly higher percentage, which was probably caused by the non-specific binding of SERS nanotags on the pillars.

Table S3. Comparison of experimental and theoretical possibility of finding different molecule numbers on pillar array.

Molecule number	1	2	3	4
Experimental probability (%)	11.52±0.37	0.83±0.041	0.036±0.0073	0.0206±0.0073
Theoretical probability (%)	9.05	0.45	0.015	0.0004

The experimental data is based on three technical replicates on three chips with the data showing mean ± S.D..

In the light of reviewer 2's comment, we have now clarified the multiplex detection detail and included the Poisson distribution comparison (Table S3) in the revised manuscript on page 13 as follows.

“As the targets independently follow Poisson distribution, each of the cytokine was separately controlled to activate less than 10% pillars.”

“As the four SERS nanotags provide distinguishable signals, we calculated the experimental Poisson distribution of finding 1-4 molecules on a single pillar. The probability of having more than one cytokine molecule per pillar is very low (<1%) by controlling the active pillars within 10%. Compared to the theoretical Poisson distribution, the experiment reported a close but higher value (Table S3), which was probably due to minor non-specific binding of SERS nanotags on the pillars.”

For the melanoma samples, all the measured cytokines were fed into an LDA algorithm. However, based on the results presented in Figure 7E, only the 1st latent variable provided discrimination between stable disease and elevated cytokine levels. Were all 4 cytokine values useful for this discrimination? Did this same classification model perform well for the other grade 2 or 3 patient samples?

Reply: To study the necessity of the four cytokines for discrimination, we now further conducted LDA with the use of two or three types of cytokines of the patient in Figure 7E. Figure S16 compares the representative LDA result utilising two, three, and four cytokines. By

including all cytokines in LDA, the discrimination accuracy improved in comparison to the use of two and three cytokines, which indicated the important role of using the four targets.

Figure S16. LDA of patient 1 who developed severe irAEs with (a) two cytokines (FGF-2 and GM-CSF), (b) three cytokines (FGF-2, GM-CSF, and CX3CL1), and (c) four cytokines (FGF-2, G-CSF, GM-CSF, and CX3CL1).

As for other patient sample classification, we have now applied LDA on all samples and integrated the data in Figure S17 and S18. For the patients who developed severe irAEs during immune checkpoint blockade treatment, LDA can classify the time point into separate zones to identify irAEs (Patient 2-5). In contrast, for patient 7 and 8 who showed mild or no sign of irAEs, LDA analysis failed to separate the data into different zones. Furthermore, for Patient 9 and 10 who developed mild irAEs (grade 1), LDA showed the discrimination. Thus, the established LDA model for the four cytokines can provide an approach in identifying severe irAEs for monitoring immune checkpoint blockade therapy.

Figure S17. Digital nanopillar SERS assay for monitoring melanoma patients who developed irAEs during immune checkpoint therapy. The average cytokine concentration graph (a, c, e, and g) the longer horizontal lines denote the median and the shorter horizontal lines denote the interquartile ranges; and corresponding LDA analysis (b, d, f, and h), respectively. IPI = ipilimumab, PEMBRO = pembrolizumab; G3=grade 3, G2=grade 2; SD=stable disease, PR=partial response. * $p < 0.05$, ** $p < 0.01$, **** $p < 0.0001$.

Figure S18. Digital nanopillar SERS assay for monitoring melanoma patients who had not developed severe irAEs during immune checkpoint therapy. The average cytokine concentration graph (a, c, e, and g), the longer horizontal lines denote the median and the shorter horizontal lines denote the interquartile ranges; and corresponding LDA analysis (b, d, f, and h), respectively. IPI = ipilimumab, PEMBRO = pembrolizumab; G3=grade 3, G2=grade 2; SD=stable disease, PR=partial response. * $p < 0.05$, ** $p < 0.01$, *** $p < 0.0001$.

As suggested by reviewer 2, we have now included the discussion on the role of four cytokines in LDA (Figure S16), as well provided the LDA of more patient samples (Figure S17, 18) in the revised manuscript on page 16 as follows.

“We further demonstrated LDA with the use of two and three cytokines for the patient classification (Figure S16), which generated less accurate results than using all four cytokines for discrimination.”

“For these severe irAEs patients, the LDA model showed a clear discrimination in cytokine profile and this could help to identify patients at risk of irAEs (Figure S17).”

“LDA was not able to classify patient 7 and 8 who had mild irAEs and didn’t show irAEs, but it recognised the minor difference in patient 9 and 10 who showed grade 1 irAEs (Figure S18).”

The authors discuss that similar cytokine quantification approaches have been used for irAEs but looking at relative values over a dynamic process. However, the authors have not provided a robust quantification of the healthy variation of these cytokines nor a threshold for determining what constitutes elevated levels, hence the information presented here is ultimately a relative metric itself. Further analysis of the recovery of individual cytokines in human serum samples is needed before the presented data can convincingly extend beyond relative values.

Reply: We believe that there might be a misunderstanding. In our work, the “relative” measurement is related to the patients’ longitudinal change in cytokine level, which we referred to as relative change. This does not relate to the capability of our assay to quantify the cytokines. Our technology allows trace quantification of the cytokines to as low as atto-molar levels. In contrary, the reviewed methods (Mol. Autism, 2017, 8, 63; Clin. Cancer Res., 2019, 25, 1557-1563) had insufficient sensitivity to quantify the cytokines at these trace concentrations. Instead, for the cytokine with concentrations lower than the limit of detection, the cytokine levels were determined by relating to a standard that had the observed median fluorescence value closest to the median of the test sample. Therefore, the term “relative” used in these studies related to the methods capability for quantification. For our proposed method, “relative” related to the change in cytokine levels of the patient during immunotherapy.

For the healthy samples, we further tested the cytokine levels in 10 healthy human serum samples (Table S9). The cytokine concentrations in healthy human serum samples were lower than the limit of detection of ELISA kits.

Table S9. Concentration of 4 cytokines in the serum of 10 healthy people as determined by digital nanopillar SERS platform.

	FGF-2 (fM)	G-CSF (fM)	GM-CSF (fM)	CX3CL1 (fM)
H1	8.05±0.46	6.53±0.33	8.25±0.23	7.05±0.29
H2	5.77±1.04	1.82±0.55	3.29±0.57	3.90±0.96
H3	1.46±0.42	0.35±0.27	0.75±0.46	1.19±0.81
H4	7.40±1.36	2.15±1.09	5.87±1.09	4.48±1.38
H5	2.93±0.89	1.13±0.36	2.04±1.18	1.68±0.96
H6	2.95±0.69	1.73±0.25	2.24±0.86	1.19±0.58
H7	4.89±1.06	1.00±0.44	3.53±0.55	3.04±0.92
H8	8.59±1.05	1.57±0.67	4.69±0.40	3.16±0.32
H9	5.94±0.99	4.85±1.29	6.71±0.99	1.22±0.07
H10	1.46±0.54	1.94±0.95	1.66±0.72	1.23±0.57

The experimental data is based on three technical replicates on three chips with the data showing mean ± S.D.

We have now clarified the relative quantification and included the results for the cytokine concentration in healthy samples (Table S9) into the revised manuscript on page 18 and 15, respectively, as follows.

“As for relative quantification (Mol. Autism, 2017, 8, 63; Clin. Cancer Res., 2019, 25, 1557-1563), the cytokine concentrations are determined by relating to a standard that had the observed median fluorescence value closest to the median of the test sample. The relative concentration, however, may fail to represent accurate cytokine levels and thus needs further exploration.”

“Following the demonstration of the accuracy of digital nanopillar SERS platform, we tested the four cytokine levels in 10 healthy people (Table S9). These 10 healthy people showed cytokine concentrations beyond the conventional ELISA capability to accurately quantify, which was consistent with previous reports (Clin. Cancer Res., 2019, 25, 1557-1563; Proc. Natl. Acad. Sci., 2019, 116, 4489-4495; Analyst, 2015, 140, 6277-6282).”

Specific details:

Throughout the manuscript the authors provide no quantitation of the assay counting performance for tested conditions. Median and interquartile ranges should be included, at least

in the captions, for all images to indicate the variability in the performance between different fields of view and chips imaged.

Reply: We thank the reviewer for the suggestion. We have now included the median and interquartile ranges in the captions for all SERS images in the revised manuscript.

Melanoma patients are arbitrarily numbered and should be re-ordered into groups such for clarity (ie 1,2,3,4,5 instead of 1,2,3,7,8)

Reply: Melanoma patients have now been re-ordered as consecutive numbers in the revised manuscript.

Figure 1e: digital counting box colors and counts do not match SERS map grid counts.

Reply: We thank the reviewer for pointing out the issue. Figure 1e has been revised to make the nanobox colors and counts match SERS map grid counts.

Figure 1. Digital nanopillar SERS platform for parallel counting of four types of cytokines. SEM images of (a) pillar array side view, (b) nanoboxes, and (c) a single nanobox on the top of a pillar; (d) SERS spectra of nanoboxes conjugated with DTNB, MBA, TFMBA, or MMTAA Raman reporters; (e) workflow for multiplex counting of cytokines.

Figure 7d&i, figure S10, the indication of grading and disease status are difficult to read given

they are located inconsistently throughout the axes. Include this information more uniformly, perhaps in the x axis.

Reply: As suggested by the reviewer, we have now reorganised the grading and disease status all consistently in the x axis.

Digital nanopillar SERS Profiling of cytokines: Nine SERS images were acquired for each sample with 60x48 micron dimensions - non overlapping? were these averaged or used as individual data points?

Reply: For each sample on a chip, we took nine non-overlapping SERS images and each image had a 60 μm × 48 μm dimension (i.e. total scanning area 9 × 60 μm × 48 μm). The counting numbers in the nine images were added together in quantifying target concentration.

We have now clarified this in the experimental section in the revised manuscript on page 23 as follows.

“For each pillar array, nine separate scanning areas (non-overlapping) were taken in total and the total active pillars were used for quantification.”

Instrumentation: Raman system was calibrated against silicon first-order 520 cm^{-1} peak, not 520 nm.

Reply: We thank the reviewer for pointing out this typo. We have now corrected it in the instrumental section of revised manuscript.

Data analysis: Log transforming counting data is appropriate for parametric testing however, true Poisson distributed events have a correlation between mean and variance. Therefore, the Kruskal Wallis test, assuming unequal variances is likely more appropriate for this data set of 10 patients. Furthermore, multiple comparison corrections should be made in order to control for errors appropriately. Finally, the pairwise comparisons indicated for time points in the melanoma samples are unclear; perhaps using colors to indicate comparison of interest would help.

Reply: We agree with the reviewer. Based on the assumption of unequal variance, the statistical test of 10 melanoma patients have now been re-conducted with the Kruskal Wallis

test among three groups and Mann-Whitney test between two groups. The data has been updated and shown in Figure S17 and S18.

To control the error appropriately, we performed multiple comparisons using Dunn's test in statistical analysis.

All the pairwise comparisons have used color-coding to indicate the target for clarity.

We have now updated the data with new tests, used colour comparison, and clarified the data analysis in the revised manuscript on page 23 as follows.

“Statistical analysis assuming unequal variances were conducted with Kruskal Wallis test among three groups or Mann-Whitney test between two groups with GraphPad Prism 8.4. To control the error appropriately, we performed multiple comparisons using Dunn's test.”

Figure S7f, was the linear fit constrained to include the 0aM data? The data point for 2.6aM does not match the value reported in the text (log 0.35% should be approx -0.6).

Reply: We thank the reviewer for pointing out this oversight. We didn't include 0 aM data in the linear curve. The data point for 2.6 aM resulted in 0.62 % active pillars, which fits the -0.21 value in the linear curve. We accidentally used the active pillar percentage of FGF-2 detection in the multiplexed assay.

We have now doubled-checked all the active percentage of pillars in the original Figure S7f and made the corrections in the revised manuscript on page 12.

Define the meaning of the error bars throughout the entire work.

Reply: The error bars represented the standard deviation from three independent technical measurements on three chips.

We have now clarified this point throughout the entire revised manuscript.

REVIEWER COMMENTS

Reviewer #1 (Remarks to the Author):

Comments:

For questions I raised last time, some of them need to be considered carefully again. Overall, this manuscript could be accepted for publication after a minor revision. My detailed comments are as follows:

1. The author still not discussed the distribution of the four kinds of antibodies. How can you demonstrate that the uniform distribution of the four kinds of antibodies on the same pillar?
2. As the author said that commercial ELISA kits couldn't generate detectable signals due to the insufficient detection sensitivity, the gold standard for clinic cytokine detection should be compared with this SERS method, how about the MALDI-TOF MS method as mentioned in this section?
3. As the author said, the sensitivity of this method is comparable with reported electrochemical luminescence methods, the advantages of this SERS method should be highlighted.

Reviewer #2 (Remarks to the Author):

Review of NCOMMS-20-21281-T

Title: A novel, disruptive, single molecule digital nanopillar SERS platform for the prediction and monitoring of immune related toxicities in immune checkpoint blockade therapy

The authors have addressed many of the posed questions during revision, leading to an improved manuscript. However, a few points require clarification for a broader audience.

1. Nanobox SNR for counting. Figure S6 depicts a fairly large variation in the intensity of the 1330cm⁻¹ peak relative to the background and also the noise fluctuations. Instead of 9 representative spectra from individual nanoboxes, the authors should report descriptive values (median, IQR, etc) for the peaks used for counting when a nanobox is present and when absent. The authors should also characterize the SNR of the peaks used for counting relative to the standard deviation in the intensity at a location in the spectrum with no peak present.
2. The width of peaks for counting. Fig 1D indicates the pure spectra with the peaks utilized for counting. However, as individual nanoboxes the peaks are significantly weaker and broader. Given the width and noise of the individual nanobox spectrum peaks (Fig 2C, Fig 3EF, Fig S6, Fig S15), it is unclear how the authors distinguished well between 1288, 1330, and 1380cm⁻¹ features. Did they use Gaussian/Lorentzian peak fitting, did they only take the intensity at the point ignoring the shape of the feature? How did they assign SERS tag membership for MMTAA, TFMB, & DTNB when all of the intensities were approximately equal as depicted in several of the individual spectra in Fig S6?
3. The authors have added additional language to indicate that interference from sample components is not an issue.
"Figure S15 shows five representative SERS spectra on the pillars for the detection of FGF-2 in human serum. Due to the rapid scan (0.05 s Raman integration), only Raman signals on FGF-2 SERS nanotags were observed without interference from other molecules in the human serum sample."
Upon reading this statement, it was not immediately clear that this finding was intended to extend to all of the SERS tags. One interpretation, that should be clarified, could be that in a multiplex experiment, human serum interfered with collection of Raman signals from G-CSF, GM-CSF, and CX3CL1 SERS nanotags, which would significantly limit the claims for multiplex imaging. The authors should clarify the wording, or justify the value of the approach if serum is a significant limitation.
4. Why is the DTNB 1556 cm⁻¹ peak enhanced as strongly as the 1330 cm⁻¹ peak, despite differences in pure spectra? Is this a function of system spectral response/quantum efficiency or is this an actual enhancement? Spectral response correction of acquired spectra will clarify this point.

5. Despite the noted "intricacies of Raman particle mapping" the argument that small nanoboxes, cause the elongated bright spots during mapping does not seem to be a justified or sufficient explanation. As you note, the individual nanobox dimensions are below the lateral and axial resolution of the confocal Raman system, even using the objective and wavelengths described here; this should mean that a single 77nm nanobox is significantly smaller than the 1 micron spot size of the laser, and that significant spatial oversampling would have to be performed for this level of spatial aliasing. The roughly 700 nm step size between mapping points (as described in the updated methods) does not seem reasonable to account for this signal. Small particles should at most appear at the center of 4 illuminated pixels (2x2 grid) depending on placement, and not in an extended line.

6. The description of controlling the cytokine for Poisson criteria during multiplex counting is still confusing. In the updated text, the authors state:

"As the targets independently follow Poisson distribution, each of the cytokine was separately controlled to activate less than 10% pillars."

Which I interpret as individually being diluted to 25,000 of each cytokine based on the estimated 250,000 pillars per chip.

However, in the supplemental information, the description indicates otherwise.

" To study the experimental Poisson distribution, we used the chips with four cytokines and SERS nanotags to have a total of 10% active pillars (i.e., each cytokine activated 2.5% pillars)."

The paragraph added in the main text should be clarified. This sentence is particularly unclear.

"The probability of having more than one cytokine molecule per pillar is very low (<1%) by controlling the active pillars within 10%."

7. The authors have provided valuable information on the inter-chip variation for human serum samples with and without cytokines. They should comment on the significance of the the noted 2-7% variation between chips with respect to the ultrasensitive detection and the levels of differences observed between cytokine profiles for differing levels of irAE activity. For instance, a 5% variation between chips would likely not impact measurements of progression to level 3 or 4, however, using different chips for samples obtained some weeks apart that are from a mild progressing to moderate disease may face challenges for discrimination. This would be an important point to address in a wider validation with clinical samples.

8. The information presented in Figure S16 for LDA including different numbers of cytokines is not fully characterized. It is unclear if the separation of the patient 1 sample is due to the inclusion of all 4 cytokines, or if a lower number of cytokines was used with a subset including CX3CL1 and/or G-CSF. A full block design is needed to show the utility of including all 4 as opposed to the importance/influence of the final added variable.

Detailed Responses to the Reviewers' Comments

Reviewer #1

Comments:

For questions I raised last time, some of them need to be considered carefully again. Overall, this manuscript could be accepted for publication after a minor revision. My detailed comments are as follows:

Reply: We sincerely thank Reviewer 1 again for the evaluation of our manuscript and thoughtful comments. We agreed to all suggestions made by Reviewer 1 and revised the manuscript accordingly.

1. The author still not discussed the distribution of the four kinds of antibodies. How can you demonstrate that the uniform distribution of the four kinds of antibodies on the same pillar?

Reply: We thank Reviewer 1 for the question. To the best of our knowledge, we are not aware of a technology that can accurately determine the distribution of four structurally-related (same immunoglobulin G family) antibodies on a single pillar with an area of $1 \mu\text{m}^2$. Our digital assay has the advantage that it does not rely on the distribution of all four antibodies on a single pillar due to the high pillar redundancy (250,000 pillars in a chip) and excess amounts of antibodies across the chip. Even if some pillars do not uniformly carry all four kind of antibodies, the assay will still work because of the use of excess amounts of antibodies that ensure the required distribution of all four kinds of antibodies for digital analysis across the chip. This has been shown in our comprehensive specificity and sensitivity experiments (revised manuscript Figure 4 and 6, and SI Figure S9-12), where the digital read-out allows digital calibration of each cytokine.

In the light of Reviewer 1's comment, we now provide additional SERS mapping images that further indicate the required distribution of the four kinds of antibodies on the pillar array in the supporting information of the revised manuscript (Figure S3). These images were obtained from the simultaneous analysis of all four target cytokines at 1031 aM. Considering the large redundancy of pillars, selected strategy for antibody conjugation on pillar surface, spectroscopic ellipsometry and MALDI-TOF MS characterisations of the pillar array, and comprehensive specificity experiments, we can infer with high confidence that we have the required antibody distribution across the chip to enable the digital analysis.

Figure S3. Representative SERS mapping images obtained from the analysis of an equimolar cytokine mixture (1031 aM). The signal distribution of the SERS nanotags indicate a required conjugation of anti-FGF-2, anti-GM-CSF, anti-G-CSF, and anti-CX3CL1 antibodies to the pillar array.

As suggested by Reviewer 1, we have now discussed the distribution of the four kinds of antibodies on the pillar array and Figure S3 in the revised manuscript on page 5 as follows.

“By using specific gold-thiol chemistry with the linker molecule dithiobis (succinimidyl propionate) (DSP), the gold-topped pillars were selectively functionalised with target recognition antibodies (anti-FGF-2, anti-G-CSF, anti-GM-CSF, and anti-CX3CL1) and acted as the small compartments to capture and confine the individual cytokine. Upon DSP binding on the gold-topped pillars through gold-thiol bond, DSP uses *N*-hydroxysuccinimide (NHS) ester to react with the amine groups of the antibodies (ACS Sens., 2018, 3, 2303-2310; Talanta, 2019, 203, 274-279). The successful antibody conjugation on gold-topped pillar surfaces was confirmed by matrix assisted laser desorption ionization-time of flight mass spectrometry (MALDI-TOF MS) (Figure S2), which showed high molecular weight fragments derived from antibodies. Furthermore, spectroscopic ellipsometry was utilised to estimate the antibody density on pillar surfaces. Based on the obtained film thickness of 18.5 nm, the calculated antibody surface density was 5.5 mg/m² using the Cuypers model (J. Biol. Chem., 1983, 258, 2426-2431), which was in agreement with the reported antibody density on substrate surfaces (Appl. Surf. Sci., 2020, 518, 146269). Though these characterisations indicated the presence

of antibodies on the pillar array, it was not possible to assess the exact distribution of the four structurally-related (same immunoglobulin G family) antibodies on a single pillar with an area of $1 \mu\text{m}^2$. As an advantage of the digital read-out with a large redundancy of pillars, it is not essential to have all four types of antibodies equally distributed on a single pillar for the assay to work. The combined surface area of all antibody-conjugated pillars provides an excess of cytokine binding sites, which maximises successful cytokine capture within the pillar array. Figure S3 shows the SERS mapping images of an equimolar cytokine solution (1031 aM) that provided a similar signal count for the FGF-2, GM-CSF, G-CSF, and CX3CL1 SERS nanotags, indicating a required distribution of four kinds of antibodies conjugated to the array of pillars.”

2. As the author said that commercial ELISA kits couldn't generate detectable signals due to the insufficient detection sensitivity, the gold standard for clinic cytokine detection should be compared with this SERS method, how about the MALDI-TOF MS method as mentioned in this section?

Reply: We thank Reviewer 1 for asking for a comparison of our assay with the clinical gold standard method. To the best of our knowledge, there is no gold standard method for clinical cytokine detection at attomolar levels. Commercially-available options with potential for trace cytokine analysis are the electrochemical luminescence assay by Meso Scale Discovery (as mentioned by Reviewer 1 in the previous round of revision) and single molecule ELISA Simona by Quanterix. Mass spectrometric approaches, including MALDI-TOF MS, are uncommon in the clinical setting and require multistep sample preparation and/or coupling to a chromatographic system.

As suggested by Reviewer 1 and in addition to the comparison of the above-mentioned commercial methods on page 14, we added a statement about the electrochemical luminescence and single molecule ELISA to the revised manuscript on page 14 as follows.

“Commercially available methods with potential for trace analysis of cytokines include single-molecule ELISA Simona by Quanterix and the electrochemical luminescence assay (Nature, 2018, 556, 332; Nat. Med., 2019, 25, 141-151) by Meso Scale Discovery. Compared to these two methods, the developed digital nanopillar SERS platform enabled *in-situ* multiplexed detection of four cytokines with comparable sensitivity.”

3. As the author said, the sensitivity of this method is comparable with reported electrochemical luminescence methods, the advantages of this SERS method should be highlighted.

Reply: As suggested by Reviewer 1, we have highlighted the advantages of our SERS method in the revised manuscript on page 14 as follows.

“Commercially available methods with potential for trace analysis of cytokines include the single-molecule ELISA Simona by Quanterix and electrochemical luminescence assay (Nature, 2018, 556, 332; Nat. Med., 2019, 25, 141-151) by Meso Scale Discovery. Compared to these two methods, the developed digital nanopillar SERS platform enabled *in-situ* multiplexed detection of four cytokines with comparable sensitivity. Unlike the issues of photo bleaching and poor multiplexing analysis often encountered in fluorescence (Small, 2010, 6, 2781-2795) and luminescence assays (Anal. Chim. Acta, 2018, 14-24), SERS provides the advantage of high multiplexing (e.g., 31-plex) (Chem. Mater., 2015, 27, 950-958; ChemPhysChem, 2009, 10, 1344-1354; Nat. Commun., 2018, 9, 1482) with the extremely narrow Raman linewidth and high photo stability of the Raman reporters. In addition, this digital nanopillar SERS platform can provide more accurate quantification of cytokines by greatly reducing the false positive signals with the confocal setting, thus eventually help clinicians to monitor irAEs during immune checkpoint therapy.”

Reviewer #2

Review of NCOMMS-20-21281-T

Title: A novel, disruptive, single molecule digital nanopillar SERS platform for the prediction and monitoring of immune related toxicities in immune checkpoint blockade therapy

The authors have addressed many of the posed questions during revision, leading to an improved manuscript. However, a few points require clarification for a broader audience.

Reply: We appreciate Reviewer 2 for the valuable comments that further improved the manuscript. To suit a broader audience, we have now clarified the points raised by Reviewer 2 below as well as in the revised manuscripts.

1. Nanobox SNR for counting. Figure S6 depicts a fairly large variation in the intensity of the 1330cm^{-1} peak relative to the background and also the noise fluctuations. Instead of 9 representative spectra from individual nanoboxes, the authors should report descriptive values (median, IQR, etc) for the peaks used for counting when a nanobox is present and when absent.

The authors should also characterize the SNR of the peaks used for counting relative to the standard deviation in the intensity at a location in the spectrum with no peak present.

Reply: We thank Reviewer 2 for these suggestions. Based on the acquired SERS mapping image, the median (interquartile range) of the spectral intensity at 1330 cm^{-1} in the presence and absence of nanoboxes were 183.03 a.u. (149.48-243.35 a.u.) and 18.07 a.u. (15.51-23.12 a.u.), respectively. The mean \pm standard deviation of the spectral intensity at 1330 cm^{-1} with and without nanoboxes were 213.41 ± 85.03 a.u. and 18.79 ± 6.01 a.u., respectively. As suggested by Reviewer 2, we have now replaced Figure S6 by reporting the Raman peak intensity values (median and IQR) and discussed the peak mean \pm standard deviation in the revised manuscript on page 8 as follows.

“Based on the acquired SERS mapping image, the median (interquartile range) of the DTNB peak intensity (1330 cm^{-1}) in the presence and absence of nanoboxes were 183.03 a.u. (149.48-243.35 a.u.) and 18.07 a.u. (15.51-23.12 a.u.), respectively. Furthermore, the mean \pm standard deviation of the DTNB peak intensity with nanoboxes (213.41 ± 85.03 a.u.) distinguished clearly from the position without nanoboxes (18.79 ± 6.01 a.u.), which demonstrated the feasibility of correctly identifying the presence of nanoboxes.”

2. The width of peaks for counting. Fig 1D indicates the pure spectra with the peaks utilized for counting. However, as individual nanoboxes the peaks are significantly weaker and broader. Given the width and noise of the individual nanobox spectrum peaks (Fig 2C, Fig 3EF, Fig S6, Fig S15), it is unclear how the authors distinguished well between 1288 , 1330 , and 1380cm^{-1} features. Did they use Gaussian/Lorentzian peak fitting, did they only take the intensity at the point ignoring the shape of the feature? How did they assign SERS tag membership for MMTAA, TFMBA, & DTNB when all of the intensities were approximately equal as depicted in several of the individual spectra in Fig S6?

Reply: We thank Reviewer 2 for the comment. To assign the SERS nanotag membership for DTNB, MBA, TFMBA, and MMTAA, we used the software Project Five 5.0 from WITec company by utilising different filters that can sum a certain spectral range on the acquired spectra and remove the background. Specifically, we employed four filters that summed the spectral range of 40 cm^{-1} with the centre position at the characteristic Raman peak of each reporter and subtracted the background with a polynomial algorithm. The filter ranges of the Raman reporter DTNB, MBA, TFMBA, and MMTAA coated SERS nanotags were (1310 - 1350 cm^{-1}), (1060 - 1100 cm^{-1}), (1360 - 1400 cm^{-1}), and (1268 - 1308 cm^{-1}), respectively. This

spectral processing enabled us to distinguish the four SERS nanotags. As suggested by Reviewer 2, we have now clarified the SERS nanotag identification in the experimental section of the revised manuscript on page 24 as follows.

“To assign the SERS nanotag membership for DTNB, MBA, TFMBA, and MMTAA, Project Five 5.0 software from WITec was utilised to create four filters, which summed a spectral range of 40 cm^{-1} with the centre position at the characteristic Raman peak of each reporter and subtracted the background with a polynomial algorithm. Specifically, the filter ranges of four Raman reporter DTNB, MBA, TFMBA, and MMTAA coated SERS nanotags were ($1310\text{-}1350\text{ cm}^{-1}$), ($1060\text{-}1100\text{ cm}^{-1}$), ($1360\text{-}1400\text{ cm}^{-1}$), and ($1268\text{-}1308\text{ cm}^{-1}$), respectively.”

3. The authors have added additional language to indicate that interference from sample components is not an issue. “Figure S15 shows five representative SERS spectra on the pillars for the detection of FGF-2 in human serum. Due to the rapid scan (0.05 s Raman integration), only Raman signals on FGF-2 SERS nanotags were observed without interference from other molecules in the human serum sample.” Upon reading this statement, it was not immediately clear that this finding was intended to extend to all of the SERS tags. One interpretation, that should be clarified, could be that in a multiplex experiment, human serum interfered with collection of Raman signals from G-CSF, GM-CSF, and CX3CL1 SERS nanotags, which would significantly limit the claims for multiplex imaging. The authors should clarify the wording, or justify the value of the approach if serum is a significant limitation.

Reply: We agree with Reviewer 2 and clarified the statement about the capability of the assay to detect simultaneously all four SERS nanotags in serum in the revised manuscript on page 15 as copied below.

“The rapid scan rate (i.e., 0.05 s for Raman signal integration) facilitated the detection of Raman signals from FGF-2, G-CSF, GM-CSF, and CX3CL1 SERS nanotags rather than the non-target molecules present in human serum due to their low Raman cross-section. As a representative example, Figure S15 shows the Raman signal distribution of the FGF-2 SERS nanotags on five different spots on the pillar array obtained from the recovery test without noticeable Raman signals from other molecules.”

4. Why is the DTNB 1556 cm^{-1} peak enhanced as strongly as the 1330 cm^{-1} peak, despite differences in pure spectra? Is this a function of system spectral response/quantum efficiency

or is this an actual enhancement? Spectral response correction of acquired spectra will clarify this point.

Reply: We thank Reviewer 2 for the question. We agree with Reviewer 2 that the peaks at 1556 cm^{-1} and 1330 cm^{-1} of DTNB on the pillar array show differences in comparison to the pure DTNB peak in solution. This is owing to the different localised environment of the molecule residing conditions. Specifically, the pure DTNB SERS spectra was acquired with nanoboxes in solution, which generated the ensemble signals contributed by all nanoboxes. However, for the SERS mapping detection on pillar array, the anisotropic nanoboxes took random orientations on the substrates, which may cause the variations of the peak intensity at 1556 cm^{-1} due to the orientation of particles relative to the laser polarisation (Nano Lett., 2007, 7, 1013-1017). For instance, Figure S15 below has three spectra (red, purple, and green lines) showing that the 1556 cm^{-1} peak is lower than 1330 cm^{-1} peak and two spectra (black and blue lines) displaying similar intensities of both peaks. As requested by Reviewer 2, we have clarified this point in the revised manuscript on page 15 as follows.

Figure S15. Representative SERS spectra of FGF-2 SERS nanotags in detecting FGF-2 in human serum. The two labelled peaks are from DTNB reporters in FGF-2 SERS nanotags.

“It is worth noting that unlike the solution-based DTNB labelled SERS nanotag spectra in Figure 1d, some of the peaks at 1556 cm^{-1} and 1330 cm^{-1} in Figure S15 had a similar intensity,

which was probably because of the different orientation of the anisotropic nanoboxes on the substrate relative to the polarisation of excitation laser (Nano Lett., 2007, 7, 1013-1017).”

5. Despite the noted "intricacies of Raman particle mapping" the argument that small nanoboxes, cause the elongated bright spots during mapping does not seem to be a justified or sufficient explanation. As you note, the individual nanobox dimensions are below the lateral and axial resolution of the confocal Raman system, even using the objective and wavelengths described here; this should mean that a single 77nm nanobox is significantly smaller than the 1 micron spot size of the laser, and that significant spatial oversampling would have to be performed for this level of spatial aliasing. The roughly 700 nm step size between mapping points (as described in the updated methods) does not seem reasonable to account for this signal. Small particles should at most appear at the center of 4 illuminated pixels (2x2 grid) depending on placement, and not in an extended line.

Reply: We thank Reviewer 2 for this insightful comment. We agree with the reviewer that small particles should appear at the centre of 4 illuminated pixels instead of the elongated shape. The elongated spot was more likely due to several particles in close proximity that aggregated on the substrate, which can happen during sample preparation processes (e.g., centrifugation) (Phys. Chem. Chem. Phys., 2015, 17, 21120-21126). However, it was difficult to visually resolve individual nanoboxes from nanobox aggregates in the SEM image (Figure 2a). Nevertheless, unlike the intensity-based assay, the aggregation of nanoboxes will not skew the digital readout result, because each cytokine will occupy a single pillar following Poisson distribution and both aggregated and individual nanoparticles were regarded as a single binding event that truly reflects the target number (Anal. Chem., 2019, 91, 9435-9441). In the light of Reviewer 2’s comment, we have now clarified this point in the revised manuscript on page 7 and correspondingly modified Figure 2.

“The elongated bright Raman spots in the SERS mapping image were probably caused by the slight aggregation of several nanoboxes during sample preparation processes (e.g., centrifugation) (Phys. Chem. Chem. Phys., 2015, 17, 21120-21126), which was difficult to visually resolve in the SEM image (Figure 2a). However, unlike the intensity-based assay, the aggregated nanoboxes as SERS nanotags to target cytokine will not skew the digital readout result, because each cytokine will occupy a single pillar following Poisson distribution and both aggregated and individual nanoparticles are regarded as a single binding event that truly reflects the target number (Anal. Chem., 2019, 91, 9435-9441).”

6. The description of controlling the cytokine for Poisson criteria during multiplex counting is still confusing. (i) In the updated text, the authors state: "As the targets independently follow Poisson distribution, each of the cytokine was separately controlled to activate less than 10% pillars." Which I interpret as individually being diluted to 25,000 of each cytokine based on the estimated 250,000 pillars per chip. (ii) However, in the supplemental information, the description indicates otherwise. "To study the experimental Poisson distribution, we used the chips with four cytokines and SERS nanotags to have a total of 10% active pillars (i.e., each cytokine activated 2.5% pillars)." The paragraph added in the main text should be clarified. This sentence is particularly unclear. "The probability of having more than one cytokine molecule per pillar is very low (<1%) by controlling the active pillars within 10%."

Reply: (i) As correctly interpreted by Reviewer 2, each cytokine in the sensitivity study was diluted to 25,000 molecules or less (2.6 aM to 1031 aM) to determine the detection range of the assay.

(ii) In the supplemental information, we studied the probability of each pillar being occupied by different molecule numbers. This study is different from the sensitivity study in (i), as we aimed to determine the molecule numbers bound on a single pillar using the four SERS nanotags under a fixed cytokine concentration. We prepared a cytokine mixture that contained all four target cytokines at equal concentration (i.e., ~6250 molecules per cytokine, ~25,000 molecules in total). We then counted the molecule numbers on a single pillar based on the characteristic Raman peaks. As shown in Table S3, our experimental data was consistent with the theoretical Poisson distribution. In the light of Reviewer 2's comment, we have now clarified this point in the revised manuscript on page 13 as follows.

"At a cytokine to pillar ratio of 1:10, we studied the probability of each pillar being occupied by different molecule numbers. To experimentally investigate the number of molecules on a single pillar, we analysed a cytokine mixture that contained all four target cytokines at equal concentration (i.e., ~6250 molecules per cytokine). To visualise and count molecule binding events on a single pillar, we labelled the captured cytokines with the four SERS nanotags that provide clearly distinguishable signals. Under Poisson distribution, the likelihood of having two or more molecules on a single pillar is <0.45% (Table S3), which underlies the digital counting principle (Nat. Biotechnol., 2010, 28, 595-599). Compared to the theoretical Poisson

distribution, the experiment data reported a close but slightly higher value, which was probably due minor non-specific binding of SERS nanotags on the pillars.”

7. The authors have provided valuable information on the inter-chip variation for human serum samples with and without cytokines. They should comment on the significance of the noted 2-7% variation between chips with respect to the ultrasensitive detection and the levels of differences observed between cytokine profiles for differing levels of irAE activity. For instance, a 5% variation between chips would likely not impact measurements of progression to level 3 or 4, however, using different chips for samples obtained some weeks apart that are from a mild progressing to moderate disease may face challenges for discrimination. This would be an important point to address in a wider validation with clinical samples.

Reply: We thank Reviewer 2 for the suggestion. In light of the reviewer’s comments, we have added a statement about the significance of the inter-chip variation to the revised manuscript on page 15.

“Overall, the observed inter-chip variation should enable accurate identification of disease progression to severe irAEs (e.g., grade 3 or 4), but may encounter some challenges in discriminating mild progressing to moderate irAEs (e.g., grade 1 or 2).”

8. The information presented in Figure S16 for LDA including different numbers of cytokines is not fully characterized. It is unclear if the separation of the patient 1 sample is due to the inclusion of all 4 cytokines, or if a lower number of cytokines was used with a subset including CX3CL1 and/or G-CSF. A full block design is needed to show the utility of including all 4 as opposed to the importance/influence of the final added variable.

Reply: We thank Reviewer 2 for the comment. We have now performed LDA with a full block design by using all the possible combinations of two and three cytokines (Figure S16-S18 below) and compared the result with four cytokine analysis, as requested by Reviewer 2. Generally, the LDA with four cytokines showed improved classification compared to the LDA with three cytokines or less (Figure S16-17). Interestingly, the use of FGF-2/G-CSF (Figure S16a), G-CSF/GM-CSF (Figure 16e), and G-CSF/CX3CL1 (Figure S16f) generated similar separation as using the four cytokines. To further compare the classification of FGF-2/G-CSF, G-CSF/GM-CSF, G-CSF/CX3CL1, and four cytokines, we performed LDA of patient 2 (Figure S18), which suggested a better differentiation with the use of four cytokines. Therefore,

the inclusion of all four cytokines in LDA facilitated a wider and more accurate patient sample analysis. We have now clarified this point in the revised manuscript on page 17 as follows.

“We further demonstrated patient 1 LDA with the use of all combinations of two (Figure S16) and three cytokines (Figure S17). Overall, the LDA with four cytokines showed improved classification over using three or less cytokines. Interestingly, considering FGF-2/G-CSF, G-CSF/GM-CSF, and G-CSF/CX3CL1, the LDA generated similar performance to the LDA with four cytokines. To further compare the classification power of FGF-2/G-CSF, G-CSF/GM-CSF, and G-CSF/CX3CL1, and four cytokines, we performed LDA of patient 2 (Figure S18), which suggested a better differentiation with the use of four cytokines. Therefore, the inclusion of all four cytokines in LDA facilitated a wider and more accurate patient sample analysis.”

Figure S16. LDA of patient 1 who developed severe irAEs (grade 4) with the use of two cytokines.

Figure S17. LDA of patient 1 who developed severe irAEs (grade 4) with the use of three cytokines.

Figure S18. LDA of patient 2 who developed severe irAEs (grade 3) with the use of different combinations of cytokines.

REVIEWERS' COMMENTS

Reviewer #1 (Remarks to the Author):

Authors carefully addressed and revised their manuscript in response to the reviewers' comments. As a result, I think the quality of the paper has been greatly improved. Therefore, the manuscript can be accepted in its present form.

Reviewer #2 (Remarks to the Author):

The authors have carefully considered and addressed the concerns that I raised during this review process.

The changes implemented to the text and figures of the manuscript and the supplementary information have improved the clarity and interpretation of the work. The expanded details for analysis methods for spectral identification, and multivariate analysis offer clearer evaluation of the methods and related the performance to use in the specific application.

The developments of ultrasensitive cytokine detection via SERS is an interesting approach that can be applied to numerous disease entities. This platform may help stimulate further work to characterize irAEs.

I support this work for publication in Nature Communications and look forward to further developments of the digital nano-pillar SERS platform.

Detailed Responses to the Reviewers' Comments

Reviewer #1

(Remarks to the Author)

Authors carefully addressed and revised their manuscript in response to the reviewers' comments. As a result, I think the quality of the paper has been greatly improved. Therefore, the manuscript can be accepted in its present form.

Reply: We sincerely thank Reviewer 1 for this positive feedback about our work and evaluation of the manuscript.

Reviewer #2

(Remarks to the Author)

The authors have carefully considered and addressed the concerns that I raised during this review process.

The changes implemented to the text and figures of the manuscript and the supplementary information have improved the clarity and interpretation of the work. The expanded details for analysis methods for spectral identification, and multivariate analysis offer clearer evaluation of the methods and related the performance to use in the specific application.

The developments of ultrasensitive cytokine detection via SERS is an interesting approach that can be applied to numerous disease entities. This platform may help stimulate further work to characterize irAEs.

I support this work for publication in Nature Communications and look forward to further developments of the digital nano-pillar SERS platform.

Reply: We greatly appreciate the positive response from Reviewer 2 and also look forward to extending this digital nanopillar SERS platform in our future works.